# Dynamics of lineage commitment revealed by single-cell transcriptomics of differentiating embryonic stem cells

Stefan Semrau[1,2,3], Johanna E. Goldmann[2], Magali Soumillon[4,5], Tarjei S. Mikkelsen[4,5], Rudolf Jaenisch[2,6] & Alexander van Oudenaarden[1]

Gene expression heterogeneity in the pluripotent state of mouse embryonic stem cells (mESCs) has been increasingly well-characterized. In contrast, exit from pluripotency and lineage commitment have not been studied systematically at the single-cell level. Here we measure the gene expression dynamics of retinoic acid driven mESC differentiation from pluripotency to lineage commitment, using an unbiased single-cell transcriptomics approach. We find that the exit from pluripotency marks the start of a lineage transition as well as a transient phase of increased susceptibility to lineage specifying signals. Our study reveals several transcriptional signatures of this phase, including a sharp increase of gene expression variability and sequential expression of two classes of transcriptional regulators. In summary, we provide a comprehensive analysis of the exit from pluripotency and lineage commitment at the single cell level, a potential stepping stone to improved lineage manipulation through timing of differentiation cues.

[1] Hubrecht Institute–KNAW (Royal Netherlands Academy of Arts and Sciences) and University Medical Center Utrecht, Uppsalalaan 8, 3584 CT Utrecht, The Netherlands. [2] Whitehead Institute for Biomedical Research, 9 Cambridge Center, Cambridge, MA 02142, USA. [3] Leiden Institute of Physics, Einsteinweg 55, 2333 CC Leiden, The Netherlands. [4] Broad Institute, 415 Main St, Cambridge, MA 02142, USA. [5] Harvard Stem Cell Institute and Department of Stem Cell and Regenerative Biology, Harvard University, 7 Divinity Ave, Cambridge, MA 02138, USA. [6] Department of Biology, Massachusetts Institute of Technology, 31 Ames St, Cambridge, MA 02142, USA. Stefan Semrau, Johanna Goldmann contributed equally to this work. Rudolf Jaenisch, Alexander van Oudenaarden jointly supervised this work. Correspondence and requests for materials should be addressed to S.S. (email: semrau@physics.leidenuniv.nl)

In vitro differentiation is a key technology to enable the use of embryonic and induced pluripotent stem cells as disease models and for therapeutic applications[1, 2]. Existing directed differentiation protocols, which have been gleaned from in vivo development, are laborious and produce heterogeneous cell populations[3]. Protocol optimization typically requires costly and time-consuming trial-and-error experiments. To be able to design more efficient and specific differentiation regimens in a systematic way it will be necessary to gain a better understanding of the decision-making process that underlies the generation of cell type diversity[4].

Lineage decision-making is fundamentally a single-cell process[5] and the response to lineage specifying signals depends on the state of the individual cell. A substantial body of work has revealed lineage biases related to, for example, cell cycle phase or pre-existing subpopulations in the pluripotent state[4, 6–8]. The commitment of pluripotent cells to a particular lineage, on the other hand, has not yet been studied systematically at the single-cell level. We consider a cell to be committed, if its state cannot be reverted by removal of the lineage specifying signal.

Here we set out to characterize the single-cell gene expression dynamics of differentiation, from exit from pluripotency to lineage commitment. Using single-cell transcriptomics we find that retinoic acid drives the differentiation of mouse embryonic stem cells to neuroectoderm—and extraembryonic endoderm—like cells. Between 24 h and 48 h of retinoic acid exposure, cells exit from pluripotency and their gene expression profiles gradually diverge. By pseudotime ordering we reveal a transient post-implantation epiblast-like state. We also study the influence of the external signaling environment and identify a phase of high susceptibility to MAPK/Erk signaling around the exit from pluripotency. We employ a minimal gene regulatory network model to recapitulate the dynamics of the lineage response to signaling inputs. Finally, we identify two classes of transcription factors which have likely distinct roles in the lineage decision-making process.

## Results

**Retinoic acid driven lineage transition.** Mouse embryonic stem cells (mESCs) are a well-characterized model system to study in vitro differentiation. Here, we focused on mESC differentiation driven by all-trans retinoic acid (RA), which is widely used in in vitro differentiation assays[9] and has important functions in embryonic development[10]. E14 mESCs were grown feeder free in 2i medium[11] plus LIF (2i/L) for several passages to minimize heterogeneity before differentiation in the basal medium (N2B27 medium) and RA (Fig. 1a). Within 96 h the cells underwent a profound change in morphology from tight, round, homogeneous colonies to strongly adherent, morphologically heterogeneous cells (Fig. 1a). To characterize the differentiation process at the population level we first measured gene expression by bulk RNA-seq at 10 time points during 96 h of continuous RA exposure (Supplementary Fig. 1). Genes that are absent in the pluripotent state but upregulated during differentiation can reveal the identity of differentiated cell types. To find such genes we clustered all genes by their temporal gene expression profiles using k-means clustering (Methods, Supplementary Fig. 1a). By testing for reproducibility through repeated clustering (stability analysis[12], see Methods) we determined that there were 6 robust gene clusters. The two clusters that showed a continuous increase in expression over the time course (clusters 5 and 6 in Supplementary Fig. 1a), were enriched with genes that have functions in development and differentiation (Supplementary Fig. 1b). In particular, established neuroectoderm and extraembryonic endoderm (XEN) markers belonged to these clusters.

Mesodermal markers, on the other hand, were not up-regulated. (Supplementary Fig. 1c, d). This observation is in agreement with earlier reports showing that RA induces neuroectodermal and XEN lineages while suppressing mesodermal gene expression[10, 13, 14].

We next set out to identify the final cell types present after 96 h of RA exposure. The up-regulation of both ectodermal and XEN markers seemed to indicate that cells adopted these two fates. Since population level measurements are not able to resolve population heterogeneity, we turned to the recently developed

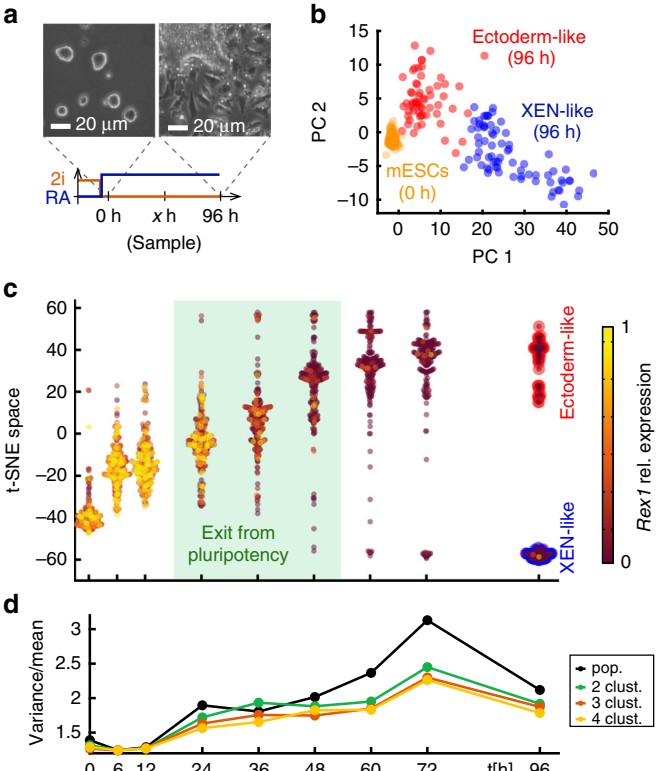

**Fig. 1** Single-cell RNA-seq revealed an RA driven lineage transition of mESCs towards ectoderm- and XEN-like cells. **a** Scheme of the differentiation protocol with phase contrast images of cells growing in 2i/L (0 h) and after 96 h of exposure to 0.25 µM RA in N2B27 medium. **b** Principal component analysis of single-cell expression profiles of mESCs and cells after 96 h of RA exposure. Principal components were calculated across all cells and time points. Cells were placed in the space of the first two principal components (PC 1 and PC 2). Each data point corresponds to a single cell. Two robust clusters identified by k-means clustering and stability analysis are shown in red (ectoderm) and blue (XEN), respectively. mESCs are shown in orange. **c** t-SNE mapping of single-cell expression profiles. The single-cell RNA-seq data (SCRB-seq) for all cells and time points were mapped on a one-dimensional t-SNE space, which preserved local similarity between expression profiles, while reducing dimensionality. Each data point corresponds to a single cell. Data points for individual time points are shown in violin plots to reflect relative frequency along the t-SNE axis. The color of each data point indicates Rex1 expression (relative to maximum expression across all cells). For the 96 h time point, two robust clusters (found by k-means clustering and stability analysis) are indicated with red or blue edges, respectively. **d** Single-cell gene expression variability quantified as the variance over the mean (Fano factor). The Fano factor was calculated either for the whole population or subpopulations of cells defined by k-means clustering using 2,3 or 4 clusters. Clustering was carried out repeatedly and the Fano factors obtained for separate clusterings were averaged

Single Cell RNA Barcoding and Sequencing method[15] (SCRB-seq, Supplementary Fig. 2). We quantified the transcriptional profiles of over 2000 single cells, sampled at 9 time points during differentiation, typically spaced 12 h apart. To visualize the heterogeneity of gene expression profiles and find subpopulations that emerge during differentiation, we used principal component analysis (PCA) and k-means clustering of the cells (Fig. 1b, Supplementary Fig. 3a, b). Repeated k-means clustering of the cells (stability analysis[12], see Methods) indicated that the population was homogeneous at 0 h and two robust clusters were present at the end of the differentiation time course (96 h). To reveal the identity of the two observed clusters, we turned to the composition of the first two principal components. The first principal component (PC 1) was primarily composed of established markers for the XEN lineage (*Sparc*, *Col4a1*, *Lama1*, *Dab2*), while PC 2 comprised markers of neuro-ectodermal development (*Prtg*, *Mdk*, *Fabp5*, *Cd24*) (Supplementary Fig. 3a, b). Accordingly, we identified one cluster as XEN-like and the other one as as ectoderm-like (Fig. 1b). Hierarchical clustering supported our interpretation of the PCA results (Supplementary Fig. 3c). In particular, we observed that genes from gene cluster 5 (Supplementary Fig. 1a), which includes ectoderm markers, were more broadly expressed in the ectoderm-like cells. By contrast, genes from cluster 6, which includes XEN markers, were largely restricted to XEN-like cells.

To confirm the existence of two cell types by an independent method, we next sought to find surface markers that would allow us to identify and purify the cell types. *Cd24*, which is among the genes with the highest loadings in PC2, is an established marker for neuroectodermal lineages[16]. *Pdgfra* is the earliest known marker of the primitive endoderm lineage in vivo[17]. Antibody staining of these two markers showed two well-separated subpopulations at 96 h (Supplementary Fig. 4a): an ectoderm-like subpopulation (CD24 + /PDGFRA-) and a XEN-like subpopulation (CD24−/PDGFRA + ). The frequencies of these two subpopulations were robust across multiple biological replicates (Supplementary Fig. 4b) and in accordance with the single-cell RNA-seq results. We then purified ectoderm-like and XEN-like cells after 96 h of RA exposure and cultured them in the same medium (N2B27 supplemented with EGF and FGF2). After continued culture, the two subpopulations showed markedly different morphologies (Supplementary Fig. 4c) and distinct gene expression patterns, as measured by bulk RNA-seq (Supplementary Fig. 4d, f). Ectoderm-like cells expressed neuro-ectodermal and neural crest markers and were similar in their expression profile to neural progenitor cells and neural crest cells in vivo. XEN-like cells expressed primitive endoderm markers and resembled an embryo-derived XEN cell line and yolk sac tissue. Taken together, these results provide evidence that the observed cell clusters corresponded to stable neuroectoderm-like and XEN-like cell types with likely in vivo correlates.

**Exit from pluripotency between 24 h and 48 h of RA exposure**. Having established the identity of the differentiated cell types we next sought to study the exit from pluripotency in detail. At the population level, we detected a gene expression response to differentiation conditions within only 6 h, as well as a second wave of gene expression changes between 24 h and 36 h (Supplementary Fig. 1e). While the immediate response was a direct effect of the switch to RA containing media, as evident from the upregulation of direct RA targets, we hypothesized that the second wave of changes indicated the exit from pluripotency. In support of this hypothesis we found that pluripotency markers were strongly down-regulated between 24 h and 48 h (Supplementary Fig. 1c, d). Cell morphology and cell cycle phase lengths

(Supplementary Fig. 5a–c) also changed significantly during the same time interval, in agreement with the observed expression dynamics. As a functional assay we used replating of the cells at clonal density in 2i/L medium. 90% of the cells could not grow in this selective medium anymore by 36 h of RA exposure (Supplementary Fig. 5d). Taken together, our population level gene expression measurements and functional assays suggested that cells exited pluripotency between 24 h and 48 h of RA exposure.

**Gradual divergence of gene expression profiles**. To visualize gene expression dynamics around the exit from pluripotency at the single–cell level we used t-distributed stochastic neighbor embedding[18] (t-SNE) of our SCRB-seq data set. t-SNE maps gene expression profiles to a low-dimensional space and places similar expression profiles in proximity to each other. Here we used t-SNE to map the expression profiles of individual cells throughout the time course on a single axis (Fig. 1c). We assessed the pluripotency status of individual cells by the expression level of the established pluripotency marker *Rex1*[19]. t-SNE showed that gene expression changed homogeneously throughout the population for the first 12 h of RA exposure, which was likely a direct effect of the RA containing medium. At this stage *Rex1* expression was high throughout the population. The subsequent steep increase in single-cell variability of gene expression at 24 h (Fig. 1d) indicated that gene expression profiles started to become more heterogeneous during the exit from pluripotency. Simultaneously, *Rex1* expression started to decline in a subset of cells, confirming the exit from pluripotency at the single-cell level. To pinpoint the time when distinguishable cell types first appeared during the differentiation time course, we calculated gene expression variability for individual cell clusters formed by k-means clustering (Fig. 1d), instead of the whole population. Starting at 48 h, within-cluster variability using 2 clusters was reduced compared to population variability, signifying the emergence of the two cell types. Clustering into 3 or 4 clusters did not reduce the variability much further. Taken together, t-SNE mapping and variability analysis showed that cells exited pluripotency and started to diverge in gene expression between 24 h and 48 h of RA exposure.

To further quantify the divergence of gene expression profiles we classified cells based on their similarity (Pearson correlation) with the average profiles of either mESCs at 0 h or the two differentiated cell types at 96 h (Fig. 2a–c). Cells which were more similar to a differentiated cell type than to mESCs first appeared between 24 h and 48 h of RA exposure, which matched the dynamics visible in the t-SNE map (Fig. 1c). Importantly, average expression profiles of the three classes were similar around the exit from pluripotency and only diverged more quickly afterwards (Fig. 2d). These observations suggested that the cells adopted the final cell fates only gradually, potentially via distinct transitory states.

**Initial differentiation into post-implantation epiblast**. We next wanted to zoom in further on the initial lineage decision, right after the exit from pluripotency, to reveal potential intermediate cell states. To achieve this goal, we had to remove possible obfuscating effects related to the asynchrony of differentiation. The transient coexistence of all three classes of cells (Fig. 2c) and the heterogeneous expression of *Rex1* (Fig. 1c) around the exit from pluripotency had indicated that differentiation was indeed asynchronous. Confounding effects due to asynchronous differentiation can be mitigated with the help of pseudo-temporal ordering of cells[20]. Here we defined a pseudo-time based on the Pearson correlation with mESCs or the differentiated cell types at 96 h (Fig. 3a, b). This pseudo-time thus reflects the progress of differentiation of an individual cell along the ectoderm- or XEN-

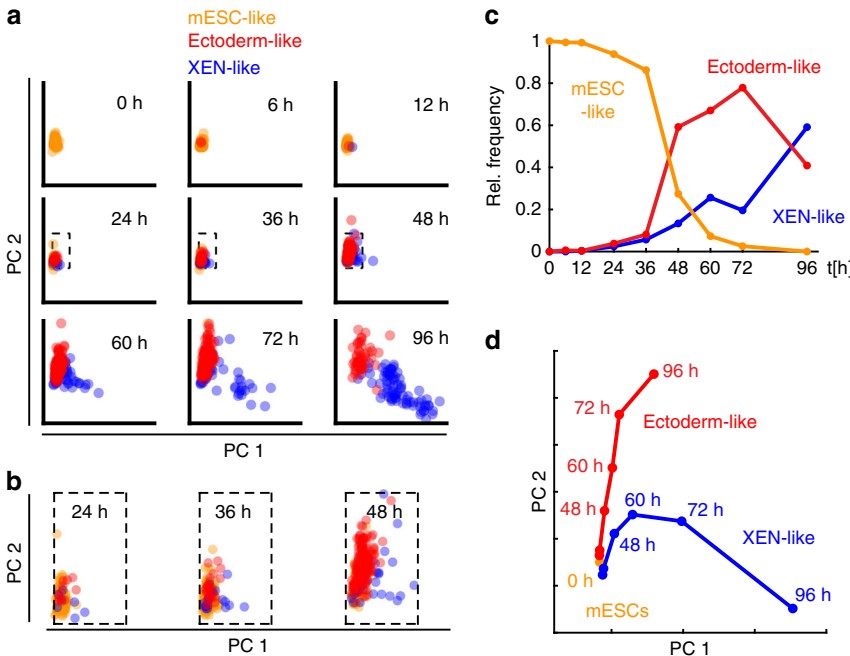

**Fig. 2** mESCs showed gradual adoption and divergence of lineage specific expression profiles. **a** Principal component analysis of single-cell expression profiles. Principal components were calculated across all cells and time points. Cells measured at the indicated periods of RA exposure were placed in the space of the first two principal components. Each data point corresponds to a single cell. Cells were classified as mESC-like (orange), ectoderm-like (red) and XEN-like (blue). Classification was based on Pearson correlation between expression profiles of individual cells and mean expression profiles of mESCs at 0 h or ectoderm-like and XEN-like cells after 96 h of RA exposure. An individual cell is identified with the cell type with which it is most strongly correlated. **b** Same data as in **a** for three select time points (24 h, 36 h and 48 h), zoomed in on the areas indicated by dashed rectangles in **a**. **c** Relative frequencies of cells classified as mESC-like, ectoderm- or XEN-like in the same way as in **a**. **d** Average movement of ectoderm- and XEN-like cells in the principal component space during RA differentiation. The positions of cells of the same type were averaged at the indicated time points

like lineage. Pseudo-temporal ordering reduced the co-existence of cell types to a small period in pseudo-time (Fig. 3c) and was thereby able to clarify expression dynamics. Furthermore, it revealed that pluripotency factors were down-regulated already before the branch point, where differentiated cell types could first be distinguished (Fig. 3d, e). During the same time, markers of post-implantation epiblast[21] (e.g. *Pou3f1*, *Fgf5*) were up-regulated. This intermediate period might represent a phase of homogeneous lineage priming or subtle population heterogeneity that we cannot resolve given the technical noise of our single-cell RNA-seq method. After the branch point, several neuroecto-dermal markers (like *Pax6*, *Sox11* or *Nes*) were up-regulated in the ectoderm-like branch. Established XEN markers (e.g. *Gata6*, *Dab2*), on the other hand, were restricted to the XEN-like branch, as to be expected.

Gene expression dynamics in pseudo-time seemed to suggest a transient state in which the cells resembled the post-implantation epiblast. To further clarify the relationship of RA differentiation with in vivo development we used PCA to compare our data set to RNA-seq measurements of pre- and peri-implantation tissues[21] (Fig. 4a and Supplementary Fig. 6). This analysis revealed that mESCs were most similar to pre-implantation epiblast (E4.5), as has been shown previously[22]. During differentiation the cells first moved closer to the E5.5 epiblast around 48 h before separating into two subpopulations (Fig. 4a). At 96 h, the XEN-like subpopulation was closest to E4.5 primitive endoderm. The occurrence of these XEN-like cells is thus likely due to a trans-differentiation from E4.5 or even E5.5 epiblast–like cells. The initial lineage decision in our system is therefore between continued differentiation along the epiblast lineage and trans-differentiation to a primitive endoderm-like state.

To confirm the single-cell RNA-seq results with an independent method we sorted cells based on PDGFRA and CD24 expression at 48 h, 72 h and 96 h and profiled the expression of the sorted subpopulations by bulk RNA-seq (Fig. 4b). At 48 h only few cells expressed PDGFRA but the majority expressed CD24. Most importantly, in PDGFRA negative cells the expression of post-implantation epiblast markers increased with CD24 expression. By 96 h the expression of post-implantation epiblast markers had largely disappeared. XEN markers, on the other hand, were expressed exclusively in PDGFRA positive cells at 72 h and 96 h. To determine cell identities in the bulk expression data set in an unbiased way we used the KeyGenes algorithm[23] together with pre- and peri-implantation tissues[21] as training set (Fig. 4c). KeyGenes identified mESCs as E4.5 epiblast, in agreement with our PCA (Fig. 4a) and previous results[22]. Notably, at 48 h PDGFRA negative/CD24 low cells were classified as E4.5 epiblast, while PDGFRA negative/CD24 high cells were identified as E5.5 epiblast. CD24 thus indicated the adoption of a post-implantation epiblast-like state, in agreement with previous findings[24]. PDGFRA positive cells, on the other hand, were consistently identified as E4.5 primitive endoderm. Bulk RNA-seq of sorted subpopulations and KeyGenes analysis thus confirmed that cells either continued to differentiate along the epiblast lineage or adopted a XEN-like cell type.

**Regulation by the external signaling environment.** Having characterized the gene expression dynamics of the exit from pluripotency and the subsequent lineage transition, we next wanted to identify effectors of the lineage decision. Notably, mESCs lost their ability to differentiate into a XEN-like lineage

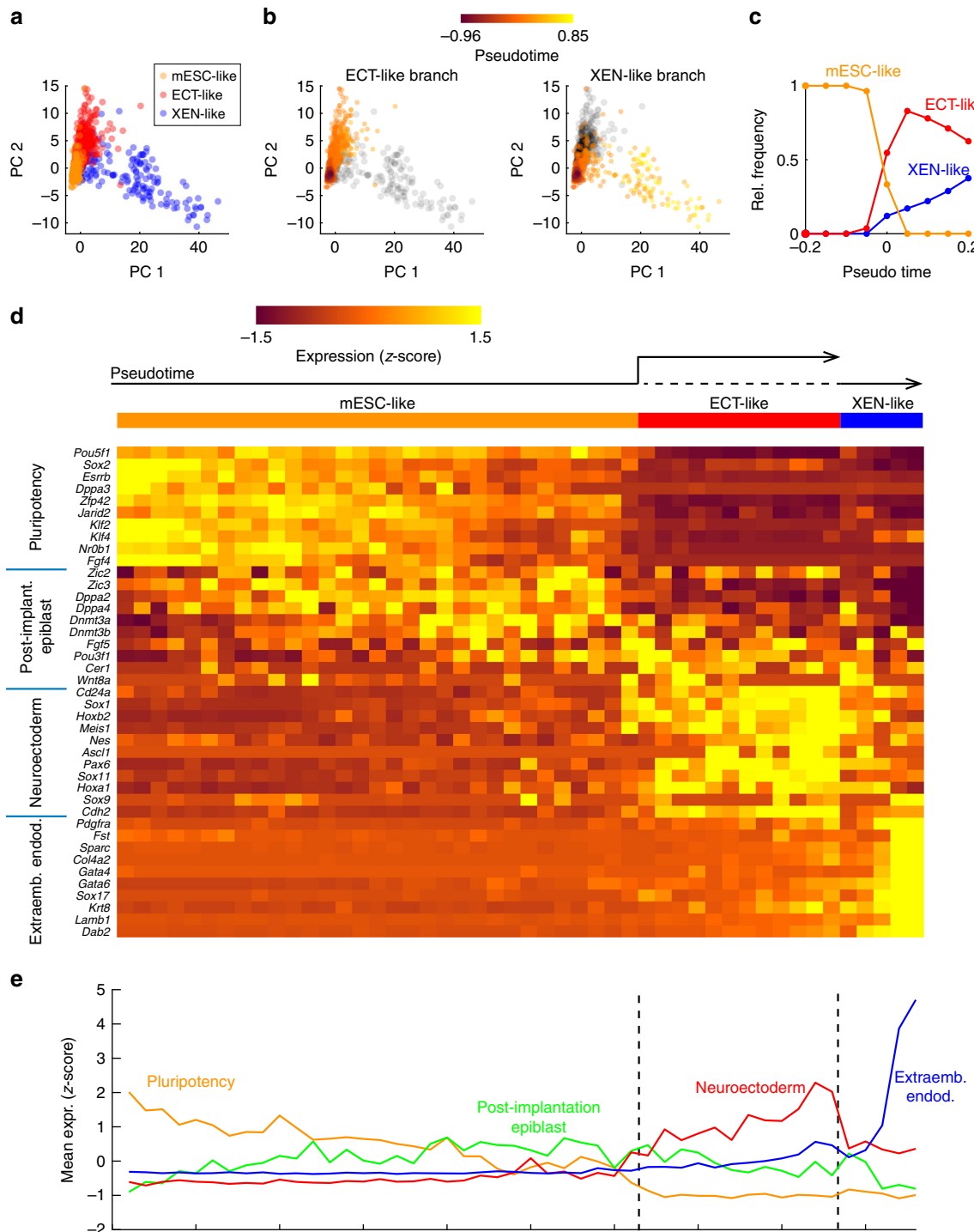

**Fig. 3** Pseudo-temporal ordering of SCRB-seq data revealed gene expression dynamics around the exit from pluripotency. **a** Classification of all cells measured during the differentiation time course. Cells were classified according to the correlation of their expression profiles with the average expression of mESCs at 0 h or ectoderm- or XEN-like cells at 96 h. Cells were placed in the space of the first two principal components (PC1 and PC2). **b** Pseudo-time of cells in the ectoderm-like branch (mESC-like cells and ectoderm-like cells, left) or the XEN-like branch (mESC-like cells and XEN-like cells, right). Pseudo-time $\tau$, which is indicated by color, is defined as $\tau = R_{pluri} - 0.5*(R_{ect} + R_{xen})$ where $R_{pluri}$, $R_{ect}$ and $R_{xen}$ are the Pearson correlations of an individual expression profile with the average expression of mESCs, ectoderm-like cells at 96 h and XEN-like cells at 96 h, respectively. **c** Relative frequencies of the three classes of cells with respect to pseudo-time. **d** Expression of a panel of marker genes for pluripotency, post-implantation epiblast[70], neuroectoderm and primitive endoderm with respect to pseudo-time. Cells were ordered by increasing pseudo-time and expression was averaged over 50 consecutive cells. Expression is presented as gene-wise z-score to accentuate temporal differences. **e** Average expression of the 4 sets of marker genes shown in **d**: pluripotency factors, post-implantation epiblast markers, neuroectoderm markers and extraembryonic endoderm markers

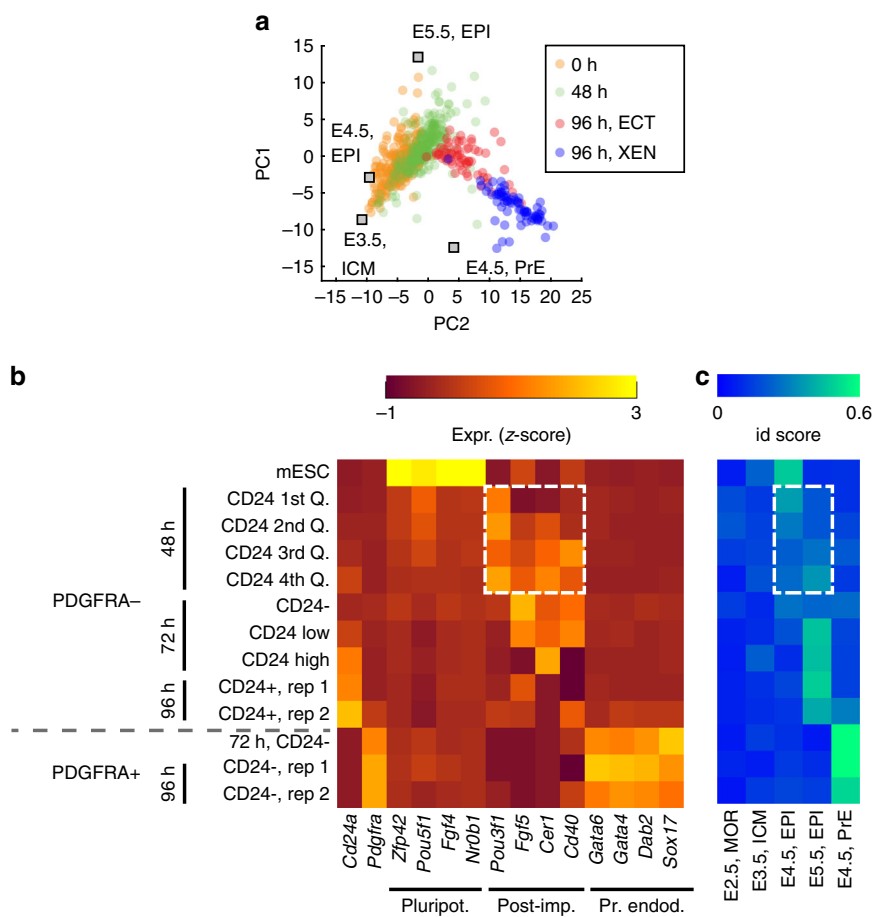

**Fig. 4** Differentiation with RA differed from the pathway observed in in vivo development. **a** Principal component analysis of a panel of pre- / peri-implantation tissues[21]. The SCRB-seq gene expression profiles obtained during RA differentiation were placed in the space of the first two principal components. Each data point represents an individual cell. Data points are colored according to duration of RA exposure and cell type (at 96 h). ICM: inner cell mass, EPI: epiblast, PrE: primitive endoderm. **b** Expression of pluripotency, post-implantation epiblast and primitive endoderm marker genes in subpopulations defined by CD24 and PDGFRA expression. Cells were sorted on PDGFRA and CD24 antibody staining by FACS before RNA extraction at the indicated periods of RA exposure (48 h, 72 h and 96 h) and bulk RNA-seq. At 48 h, PDGFRA- cells were sorted by quartiles of CD24 expression, at 72 h cells were sorted by PDGFRA expression and terciles of CD24 expression. Expression of post-implantation epiblast markers at 48 h is highlighted with a dashed white box. **c** Identity of bulk RNA-seq samples as determined by the KeyGenes algorithm[23]. A panel of pre- / peri-implantation tissues[21] was used as the training set. A high identity (id) score corresponds to a high confidence about tissue identity. (MOR: morula, ICM: inner cell mass, EPI: epiblast, PrE: primitive endoderm). The identity scores for the epiblast tissues at 48 h are highlighted by a white dashed box

when they were cultured, prior to differentiation, in serum and LIF conditions (without feeders) instead of 2i/L (Supplementary Fig. 7a, b). The ability of RA to drive ectodermal differentiation seemed unaffected under these conditions, as reported before[25]. Since culture conditions had such a strong impact on the developmental potential of mESCs we wanted to explore the contribution of specific signaling pathways on the cellular decision. We differentiated mESCs with RA in the presence of a MEK inhibitor (MEKi, PD0325901), which abrogates MAPK/Erk signaling; a GSK3 inhibitor, which effectively stimulates Wnt signaling (GSK3i, CHIR99021), LIF, which activates the JAK/Stat pathway or an FGF receptor inhibitor (FGFRi, PD173074). (Supplementary Fig. 7c–h). The first 2 of these molecules are components of the defined 2i medium and are known to prevent differentiation while stabilizing the pluripotent state. The presence of GSK3i or LIF led to an overall reduction of differentiated cells (Supplementary Fig. 7c), consistent with their role in stabilizing pluripotency. Addition of MEKi alone, however, led to a specific reduction of the XEN-like subpopulation (Supplementary Fig. 7c–e), in agreement with previous results[26, 27]. This effect was

unlikely due to interference with RA signaling since increasing RA concentration did not reverse the effect (Supplementary Fig. 7f). In contrast to the MEK inhibitor, the FGF receptor inhibitor not only suppressed the XEN-like population but also greatly reduced the ectoderm-like population (Supplementary Fig. 7g, h). This observation is in agreement with earlier studies that reported a requirement for FGF signaling in mESC differentiation[28] and lineage segregation in the early mouse blastocyst[29]. Taken together these experiments clearly demonstrate that RA driven XEN-specification requires the same signaling pathways as other differentiation regimens and XEN-specification in vivo, despite the pleiotropic nature of RA.

**Phase of high susceptibility to external signal inputs.** We next wanted to establish when mESCs are sensitive to RA signaling and how long the signal would have to be applied to drive a complete lineage transition. Having observed that gene expression responds to differentiation conditions within 6 h (Supplementary Fig. 1e), we hypothesized that a short pulse of RA might be sufficient to induce XEN specification. To test this hypothesis

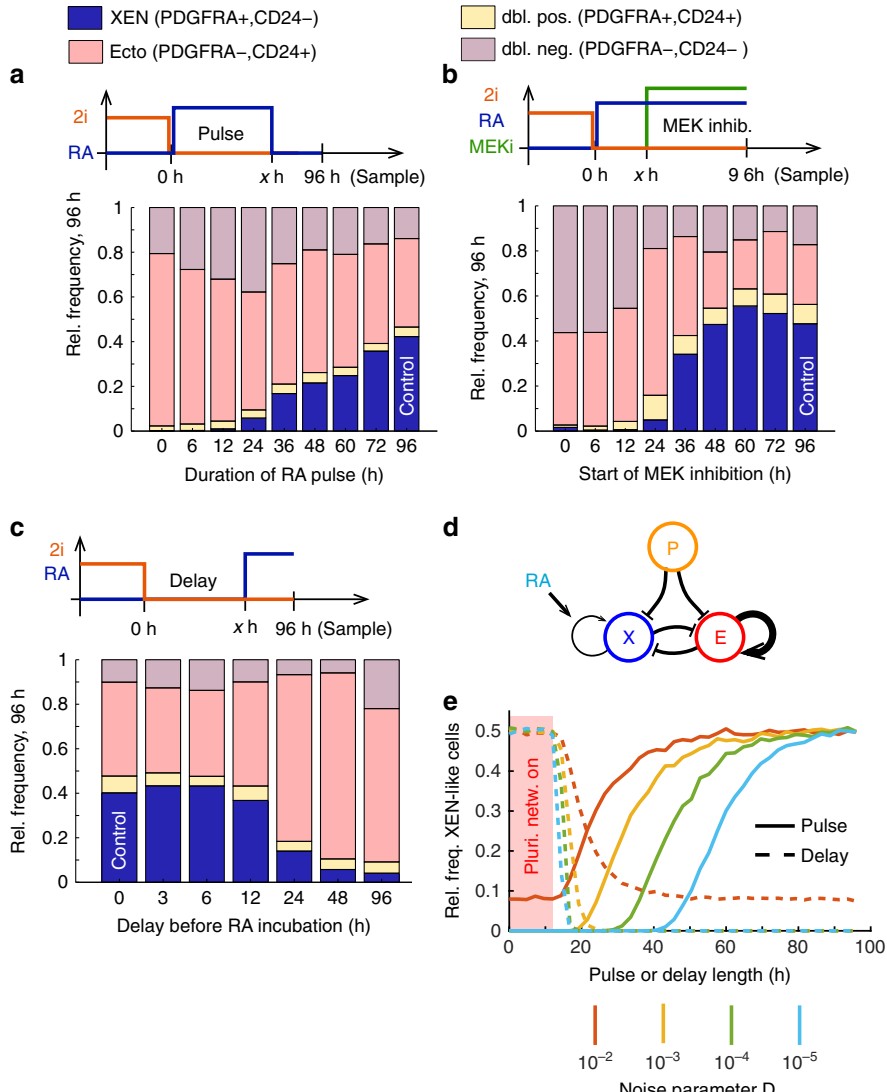

**Fig. 5** Susceptibility to signaling inputs was highly dynamic around the exit from pluripotency. **a**–**c** Fractions of cells classified as XEN-like, ectoderm-like, double positive and double negative after 96 h, based on CD24 and PDGFRA expression. Expression of the two markers was measured by antibody staining and flow cytometry. **a** Cells were pulsed with 0.25 µM RA for x h (pulse) and subsequently differentiated in basal medium (N2B27) complemented with an RA receptor antagonist. **b** Cells were incubated with 0.25 µM RA for x h (pulse) after which 0.5 µM PD0325901 (MEK inhibitor) was added for the remainder of the time course. **c** Cells were first incubated with basal medium (N2B27) for x h (delay) and then exposed to 0.25 µM RA for the remainder of the time course. **d** Schematic representation of a minimal gene regulatory network that can model a lineage decision[32]. Pointy arrows indicate (auto-) activation; blunted arrows indicate repression. E and X represent expression of ectoderm-like and XEN-like transcriptional programs, respectively. P stands for the pluripotency network. RA increases the auto-activation of the XEN program. **e** Results of the stochastic simulations of the network shown in **d**. The relative frequency of XEN-like cells after 96 h is shown vs. the length of an RA pulse (solid lines) or the length of the delay before RA exposure is started (dashed lines). In all cases the pluripotency network was turned off after 12 h. Simulations were run with different amounts of gene expression noise (D: noise power / time, see Methods). See Supplementary Fig. 8b for exemplary trajectories

we applied a precisely defined pulse of RA by first exposing the cells to RA for a defined period of time and then switching to a highly potent pan-RA receptor antagonist[30] (Fig. 5a). These experiments showed that, contrary to our expectation, RA had to be applied for at least 24 h for XEN-like cells to appear. Longer pulses resulted in a gradual increase of the XEN-like fraction. A 36 h long pulse of RA resulted in 20% XEN-like cells at the 96 h time point, roughly half of what we found after uninterrupted RA exposure (Fig. 5a). This indicated that even after 36 h of RA exposure and significant down-regulation of the pluripotency network XEN specification continued to depend on RA-signaling. Timed abrogation of MAPK/Erk signaling by MEKi resulted in a similar response as an RA pulse (Fig. 5b). At least 24 h of

uninterrupted MAPK/Erk signaling was necessary for XEN-like cells to occur. Longer durations of MAPK/Erk signaling resulted in an increase in the XEN-like subpopulation. This effect plateaued after 48 h, which suggested that XEN-like cells then became independent of MAPK/Erk signaling and thus stably committed. We also wanted to establish when cells lost their ability to respond to RA signaling. To this end we first differentiated the cells in basal (N2B27) medium and started RA exposure after a defined time period (Fig. 5c). When RA exposure was delayed by up to 12 h, we did not observe any difference in the lineage distribution at the 96 h time point. For longer delays of RA exposure, we found that the fraction of XEN-like cells declined. This observation demonstrated that the cells quickly lost

their susceptibility to RA after the exit from pluripotency. Taken together, these signaling experiments revealed a short transient phase after the exit from pluripotency, during which cells were maximally susceptible to external signaling cues to inform their lineage decision.

**A minimal gene regulatory network of lineage bias.** Interestingly, our experiments revealed a difference in the lineage response dynamics between the RA pulse and RA delay. While cells abruptly lost their ability to become XEN-like after only 12 h in N2B27 (Fig. 5c), the RA pulse had to be applied for at least 24 h to cause XEN-specification and longer pulses elicited a gradually increasing response (Fig. 5a). This asymmetry could be related to the fact that N2B27 on its own drives differentiation towards the

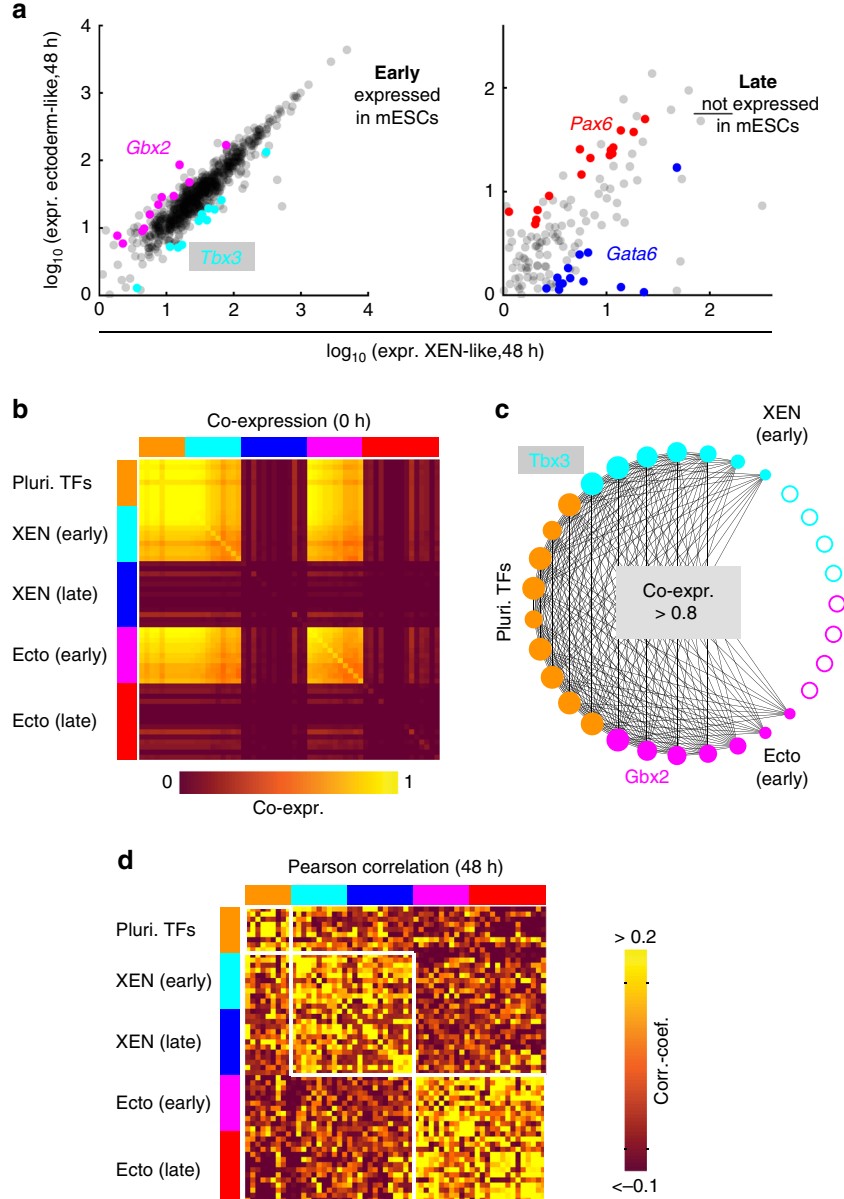

**Fig. 6** Distinct co-expression and correlation patterns identified two classes of lineage specific transcriptional regulators. **a** Expression of transcriptional regulators in ectoderm-like and XEN-like cells identified in the SMART-seq2 data set. Genes that were significantly differentially expressed after 48 h of RA exposure are shown in red or pink (overexpressed in ectoderm-like cells) and blue or cyan (overexpressed in XEN-like cells), respectively. The two panels contain genes, which are present in the pluripotent state (early, left panel) or absent in the pluripotent state (late, right panel). A list of all identified genes is given in Supplementary Fig. 9b. **b** Co-expression of transcriptional regulators in the pluripotent state. The gene set comprised the differentially expressed transcriptional regulators identified here (see **a**), as well as pluripotency related transcription factors[51] (see Supplementary Fig. 9b). Co-expression was calculated using gene expression measured by SMART-seq2. Co-expression of two genes was quantified as the fraction of cells in which the expression of both genes exceeded a certain threshold value (see Methods). **c** Co-expression network in the pluripotent state. Two genes are connected by an edge if their co-expression exceeds 0.8. The gene set comprised XEN specific regulators (cyan nodes) and ectoderm specific regulators (pink nodes) that are expressed in the pluripotent state (early factors), as well as pluripotency factors[51] (orange nodes). The radius of solid nodes is proportional to the number of connections to other nodes. Nodes without any connections are depicted as open nodes. **d** Pearson correlation between transcriptional regulators after 48 h of RA exposure. The gene set is the same as in **b**. Pearson correlation was calculated using gene expression measured by SMART-seq2

neuroectoderm lineage[31]. Correspondingly, we consistently found that the majority of cells became ectoderm-like when there was no RA present (Fig. 5a–c). To explore the role of an intrinsic epiblast or ectoderm bias we developed a simple phenomenological model based on a minimal gene regulatory network (GRN)[27, 32]. Briefly, the GRN is comprised of two lineage-specific, auto-activating expression programs that mutually repress each other (Fig. 5d, Supplementary Fig. 8a). This GRN can produce two stable attractors that correspond to two differentiated cell types. Here, we added repression of both lineages by the pluripotency network to model the pluripotent state. Consistent with our data, we assumed that the pluripotency program is turned off after 12 h. To model the ectoderm bias we assumed that auto-activation of the ectoderm program was stronger than auto-

activation of the XEN program in the absence of RA. In the presence of RA auto-activation of both programs was taken to be equal. Due to the great importance of gene expression noise in lineage decision-making[5, 4], we also incorporated noise in our model (Methods). Stochastic simulations of the 3-state GRN reproduced the asymmetry between RA pulses and delays (Fig. 5e). The frequency of XEN-like cells decreased sharply when the delay in RA signaling was extended beyond the exit from pluripotency. The RA pulse, on the other hand, had to be applied for a longer period of time to cause XEN-specification and the response was more gradual. This behavior can be explained by the fact that in the absence of RA cells are quickly drawn to the ectoderm attractor after the exit from pluripotency. When RA is added after a delay, the cells are already in the proximity of the

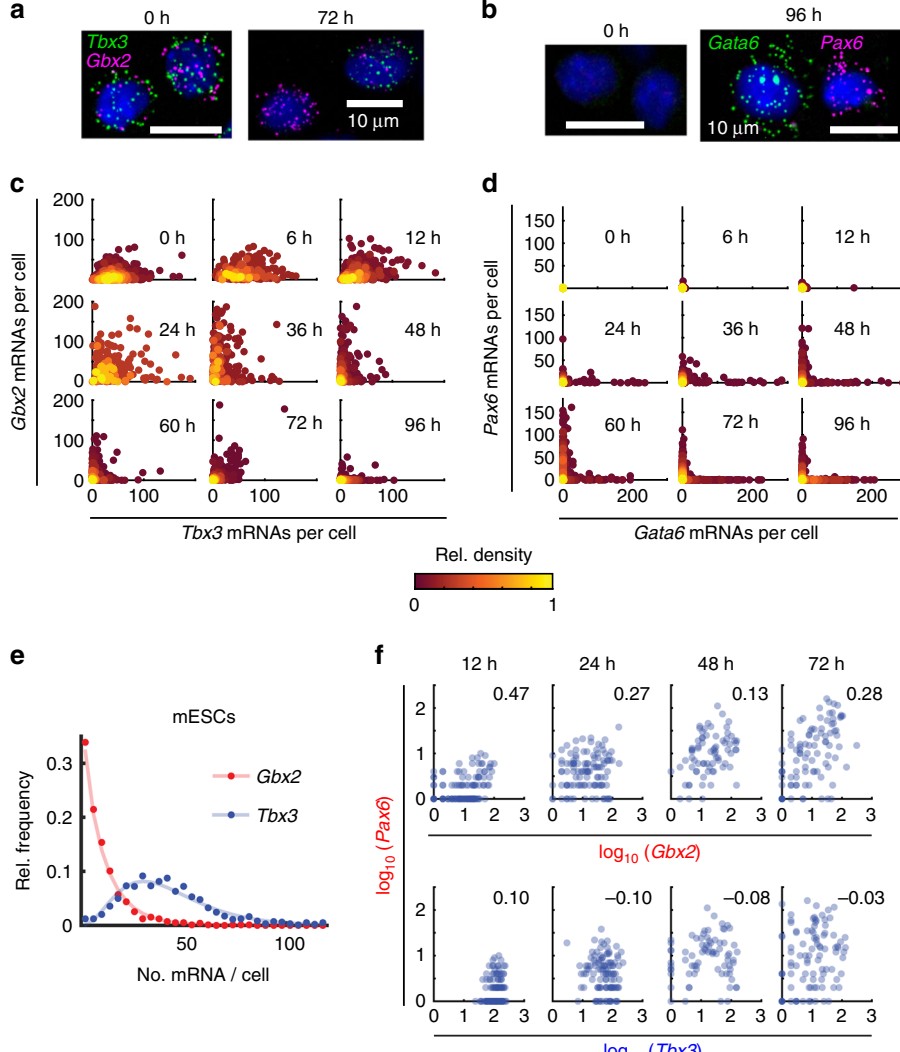

**Fig. 7** smFISH confirmed distinct expression patterns of exemplary transcription factors. **a** Fluorescence images of smFISH for *Gbx2* and *Tbx3* in mESCs (0 h) and after 72 h RA exposure. Each diffraction limited dot corresponds to a single mRNA molecule. Hoechst staining of nuclei is shown in blue. **b** Fluorescence images of smFISH for *Pax6* and *Gata6* in mESCs (0 h) and after 96 h RA exposure. Each diffraction limited dot corresponds to a single mRNA molecule. Hoechst staining of nuclei is shown in blue. **c** Scatter plots of the number of *Gbx2* and *Tbx3* mRNAs per cell measured by smFISH. Each data point is a single cell. Color indicates the local density of data points. The number of shown cells measured at a certain time point ranges between 224 and 983. **d** Scatter plots of the number of *Pax6* and *Gata6* mRNAs per cell measured by smFISH. Each data point is a single cell. Color indicates the local density of data points. The number of shown cells measured at a certain time point ranges between 293 and 570. **e** Distribution of the *Tbx3* and *Gbx2* transcripts in individual mESCs as measured by smFISH. Both data sets are fit by a Gamma distribution (*Tbx3*, $R^2 = 0.94$, solid blue line; *Gbx2*, $R^2 = 0.99$, solid red line). **f** Scatter plots of the number of mRNAs per cell for *Gbx2* and *Tbx3* vs *Pax6* measured by smFISH. Each data point is a single cell. Cells were exposed to RA for 12 h, 24 h, 48 h and 72 h, respectively, as indicated above each column of panels. The number in each panel is the Pearson correlation between the genes plotted in the respective panel

ectoderm attractor and cannot escape it anymore, which causes the lack of XEN cells. Notably, the asymmetry between the response curves was reduced by gene expression noise. Noise allowed the cells to switch between the basins of attraction of the two attractors (Supplementary Fig. 8b), thereby equalizing the intrinsic difference between the two attractors. Taken together, our stochastic simulations showed that an intrinsic ectoderm bias can explain the difference in the response dynamics between an RA signal delay and an RA pulse.

**Two classes of transcriptional regulators**. Having revealed a highly dynamic susceptibility to signaling cues, we were wondering if the expression of transcriptional regulators was equally dynamic. To that end we focused on transcriptional regulators that show lineage specific expression when the two lineages can be first discerned robustly, around 48 h (see Methods for the list of GO terms used to define transcriptional regulators). Since these regulators were typically lowly expressed, they were not well-represented in the SCRB-seq data set. Therefore, we collected another single-cell RNA-seq data set using SMART-seq2[33] at four early RA differentiation time points (0 h, 12 h, 24 h and 48 h). We first identified XEN-like and ectoderm-like cells at the 48 h time point (Supplementary Fig. 9a). The remaining cells were likely mostly undifferentiated cells as several pluripotency factors were differentially expressed in this population (Supplementary Fig. 9b). In the cells classified as XEN- or ectoderm-like we found 50 transcriptional regulators to be differentially expressed between the two lineages (Fig. 6a, Supplementary Fig. 9b). 22 of those genes (dubbed "early") were present already in mESCs. These early regulators were broadly co-expressed in individual cells at the beginning of the time course (Fig. 6b and Supplementary Fig. 9c). Compared to canonical pluripotency factors, early regulators showed a smaller level of co-expression with each other in the pluripotent state, in particular if they belonged to different lineages (Fig. 6b, c). Individual mESCs thus expressed varying ratios of XEN and ectoderm specific early regulators. Over time, co-expression of XEN and ectoderm specific early regulators declined but they never became completely mutually exclusive (Supplementary Fig. 9c). Hence, we speculated that other transcriptional regulators might be up-regulated in lineage biased cells and take over lineage specification from the early regulators. Indeed, 28 of the identified differentially expressed regulators (dubbed "late") were, by definition, not significantly expressed at the beginning of the time course (Fig. 6a, Supplementary Fig. 9b). These late regulators were overall positively correlated with early regulators of the same lineage and anti-correlated with regulators of the opposing lineage (Fig. 6d and Supplementary Fig. 9c). This correlation pattern suggested that early regulators might have a role in lineage biasing, whereas late factors could be involved in lineage commitment.

To confirm the sequential expression of early and late regulators, we next focused on four transcription factors, chosen based on their reported function for the specification of ectoderm (*Gbx2*[34] (early), *Pax6*[35] (late)) and extraembryonic endoderm (*Tbx3*[36] (early), *Gata6*[37] (late)). Notably, *Tbx3* and likely also *Gbx2* are direct targets of RA[38, 39]. In agreement with their reported roles we found these 4 factors to be differentially expressed in ectoderm-like and XEN-like cells, respectively, in our SCRB-seq data set (Supplementary Fig. 9d). To quantify correlation patterns with high precision we used single-molecule FISH (smFISH[40]) due to its superior sensitivity and precision compared to single-cell RNA-seq (Supplementary Fig. 10a). We measured the expression of the early factors (Fig. 7a, c) or the late factors (Fig. 7b, d) together with the pluripotency factor *Nanog* and quantified co-expression at all time points (Supplementary

Fig. 10b–d). In agreement with the SMART-seq2 data, early factors were broadly co-expressed in the pluripotent state and a smaller subpopulation of co-expressing cells persisted during differentiation (Fig. 7c, Supplementary Fig. 10c). Importantly, mESCs expressed the early factors at highly variable ratios: 30% of mESCs did not express the early ectoderm factor *Gbx2* at a significant level, while almost all cells expressed the early XEN factor *Tbx3* (Fig. 7e). smFISH further confirmed that late factors were only sporadically expressed before the exit from pluripotency but strongly up-regulated in separate subpopulations thereafter. These subpopulations likely corresponded to lineage-committed cell states (Fig. 7d and Supplementary Fig. 10d, e). Interestingly, a simultaneous measurement of the early ectoderm factor *Gbx2* and the late ectoderm factor *Pax6* revealed their positive correlation throughout the time course, even before the exit from pluripotency (Fig. 7f). A possible explanation for such a correlation might be a lineage-biasing role for *Gbx2*. All in all, the smFISH measurements clearly confirmed differences in the expression dynamics and correlation patterns of early and late transcriptional regulators.

## Discussion

In summary, we leveraged a recently developed high-throughput single-cell transcriptomics method to dissect the exit from pluripotency and dynamics of lineage commitment in RA driven differentiation of mESCs with high temporal resolution. We characterized the influence of the external signaling environment and explained the dynamics of the signaling response with a minimal gene regulatory network. We finally identified potential transcriptional regulators of lineage decision and commitment.

In particular, we showed that after 96 h of RA exposure mESCs had differentiated into neuroectoderm-like and XEN-like cells. By purification and continued culture we showed that these cell types are stable and not just transient expression fluctuations. In agreement with previous results[22] we found mESCs cultured in 2i/L to be transcriptionally most similar to E4.5 epiblast in vivo (Fig. 4a–c). At E4.5 the lineage decision between primitive endoderm and epiblast has already occurred, so a priori it would not be expected that mESCs should be able to generate XEN cells. The potential to create XEN-like cells could be explained by a subpopulation of cells in the pluripotent state that resembles an earlier developmental stage. In our single-cell RNA-seq data set we could not find evidence for such pre-existing heterogeneity (Fig. 4a). Alternatively, RA might have caused the dedifferentiation of the whole mESC population to an earlier developmental stage after which the cells could follow the in vivo bifurcation between E4.5 epiblast and primitive endoderm. While the whole population indeed initially moved closer to the E3.5 inner cell mass during the first 24 h, cells then moved towards E5.5 epiblast before discernible XEN-like cells appeared (Supplementary Fig. 6). Hence, most likely XEN-like cells are created by trans-differentiation from E4.5 or E5.5 epiblast-like cells and mESCs initially decide between progression along the epiblast lineage and the XEN-like cell type right after the exit from pluripotency. The epiblast lineage then further develops to neuroectoderm-like cells by 96 h. A recently published study by Klein et al. used single-cell RNA-seq to characterize mESC differentiation by LIF withdrawal[41] and also found a small XEN-like subpopulation. That and other studies[42, 27] show that XEN-like cells occur more generally in in vitro differentiation of mESCs and are not an idiosyncratic artefact of exposure to RA. We also found that mESCs grown in 2i/L (but not in serum and LIF) efficiently generate XEN cells under RA exposure (Supplementary Fig. 7a, b). Similarly, Schröter et al. have observed, for a different differentiation assay, that pre-culture in 2i/L greatly increases the

number of XEN-like cells generated from mESCs[27]. Together with those results our observations thus support a model in which mESCs grown in 2i/L functionally correspond to a slightly earlier developmental stage than mESCs grown in serum and LIF[42].

Despite the artificial nature of the lineage transition described here, we observed several similarities with the epiblast/primitive endoderm bifurcation in vivo. A recent study by Saiz et al. in the mouse embryo[43] showed that epiblast and primitive endoderm are specified asynchronously from a pool of progenitor cells, which also happened in our experiments (Fig. 2c). Furthermore, in the absence of primitive endoderm inducing signals, all cells of the inner cell mass become epiblast-like in vivo[29, 43–45]. In our experiments the majority of mESCs adopted the epiblast/ecto-dermal lineage in the absence of RA, in agreement with the literature[31, 46]. Saiz et al. observed that MEKi prevented the specification of primitive endoderm, in agreement with an earlier report by Nichols et al[47]. The experiments by Saiz et al. also revealed that the susceptibility to MEKi disappeared gradually between E3.5 and E4.5. Our experiments with MEKi showed similar dynamics (Fig. 5b). Thus, both in vivo and in vitro, cells seem to gain competence to respond to XEN specifying signals over time. In vitro, the susceptibility to signaling inputs is thought to be contingent on the down-regulation of pluripotency factors and the exit from pluripotency[46, 48], a notion which is supported by our study (Fig. 5a–c). Our results thus clearly reveal a window of opportunity right after the exit from pluripotency, which might be exploited to guide lineage decisions with maximal efficacy.

Using a minimal GRN to model the lineage decision, we also showed that an inherent epiblast/ectoderm bias can cause the observed asymmetry between an RA delay and an RA pulse (Fig. 5d). A similar GRN has been used successfully before in a report by Schröter et al., studying induction of the XEN lineage by exogenous Gata4 expression[27]. Importantly, our model does not strictly require an ectoderm bias. An initial bias for progression along the epiblast lineage (and continued differentiation to ectoderm under RA) would be sufficient. This interpretation is in line with previous results that reported expression of non-ectodermal markers during early stages of differentiation in N2B27[49]. Notably, in our model, gene expression noise was able to reduce the asymmetry between the RA regimens, because gene expression trajectories could switch more easily between the basins of attraction of the two lineage attractors (Supplementary Fig. 8b). The impact of noise in the context of lineage decisions was recently addressed in a publication by Marco et al[50]. In that study the authors focused on the ability of noise to destabilize committed cell states. Here we showed that noise can also impact commitment dynamics and even mask an intrinsic lineage bias. This result suggests that gene expression noise could be exploited to influence lineage decision-making in vitro.

Our study further identified early-expressed lineage specific transcriptional regulators that are heterogeneously expressed in the pluripotent state and thus have a potential role in biasing the lineage decision. Importantly, the two factors we studied in detail, Gbx2 and Tbx3, were previously determined to be part of an essential pluripotency network[51–55]. It has been suggested before that some pluripotency genes are also involved in lineage specification[25, 48, 49]. Thomson et al. showed that Sox2 and Oct4 promote the neuroectodermal and mesendodermal lineage, respectively[48]. Malleshaiah et al. reported similar functions for Nac1 and Tcf3, respectively[25]. Future research will have to show whether Gbx2 and Tbx3 have similar roles for the epiblast/neu-roectoderm and XEN lineage, respectively. In fact, for Tbx3 Lu et al. recently demonstrated a dual function in self-renewal and XEN specification[36]. The observed correlation between Gbx2 and Pax6 suggests a function of Gbx2 in epiblast or neuro-ectoderm specification. The long-tail distribution of Gbx2 in mESCs hints at

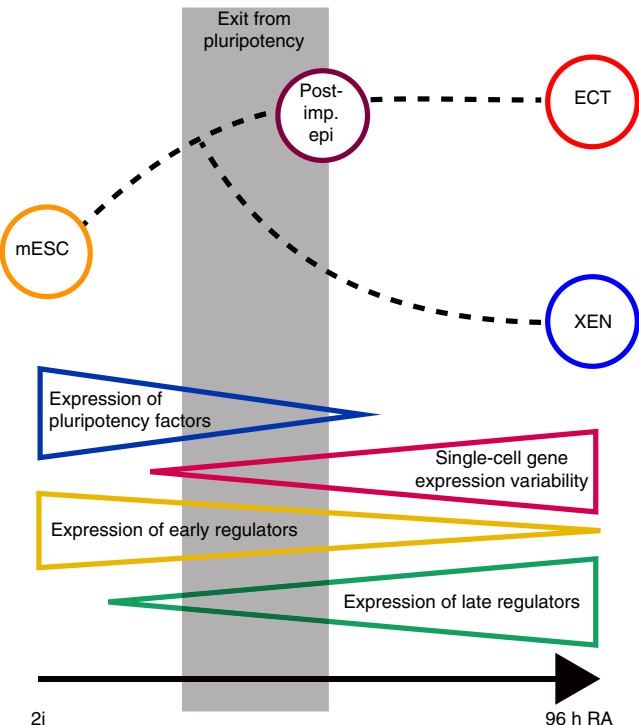

**Fig. 8** Transcriptional signatures of the lineage decision phase

infrequent transcriptional bursting and possibly distinct sub-populations[56]. The causal relationship between Gbx2 and Pax6 and the functional relevance of the Gbx2 high subpopulations will be explored in a future study. Late-expressed lineage specific transcription factors, like Pax6 and Gata6, which were not expressed in the pluripotent state, have a role in lineage com-mitment. They can thus serve as bona fide lineage markers.

Transient phases of susceptibility to lineage cues, such as the one characterized in this study, might be valuable windows of opportunity for the control of lineage decisions. We speculate that exit from a pluripotent cell state necessarily coincides with a phase of instability and increased gene expression variability, as demonstrated recently for lineage decisions in the hematopoietic system[57, 58]. Based on our results we would like to propose ten-tative transcriptional signatures of such phases (Fig. 8): 1. down-regulation of pluripotency factors (Fig. 1c), 2. a sudden increase in single-cell gene expression variability (Fig. 1d), 3. slowly diverging lineage specific expression patterns (Fig. 2d), 4. co-expression of early-expressed (thus potentially lineage-biasing) transcriptional regulators (Fig. 6b), 5. sporadic expression of late-expressed (thus potentially lineage-committing) transcriptional regulators (Fig. 7d). We hope that these results will be a stepping stone towards finding more efficient ways to guide lineage decisions.

## Methods

**Cell culture**. All cell lines were grown routinely in modified 2i medium[11] plus LIF (2i/L): DMEM/F12 (Life technologies) supplemented with 0.5x N2 supplement, 0.5x B27 supplement, 0.5mM L-glutamine (Gibco), 20 µg/ml human insulin (Sigma-Aldrich), 1 × 100U/ml penicillin/streptomycin (Gibco), 0.5x MEM Non-Essential Amino Acids (Gibco), 0.1 mM 2-Mercaptoethanol (Sigma-Aldrich), 1 µM MEK inhibitor (PD0325901, Stemgent), 3 µM GSK3 inhibitor (CHIR99021, Stemgent), 1000 U/ml mouse LIF (ESGRO). Cells were passaged every other day with Accutase (Life technologies) and replated on gelatin coated tissue culture plates (Cellstar, Greiner bio-one).

E14 cells were provided by A. van O., V6.5 cells were provided by R.J. Both cell lines were regularly tested for mycoplasma infection.

**Differentiation**. Prior to differentiation cells were grown in 2i/L for at least 2 passages, with the exception of the experiment shown in Supplementary Fig. 7a,b: here cells were grown in knockout DMEM (Thermofisher) supplemented with 10% ES cell screened FBS (Sigma), 1 × 100U/ml penicillin/streptomycin (Gibco), 0.1 mM 2-Mercaptoethanol (Sigma-Aldrich) and 1000 U/ml mouse LIF (ESGRO) for 3 passages prior to differentiation. For all differentiation experiments cells were seeded at a density of $2.5 \times 10^5$ cells per 10 cm dish and grown over night (12 h). After washing cells twice with PBS, differentiation was carried out in basal N2B27 medium (2i/L medium without the inhibitors, LIF and the additional insulin) supplemented with all-trans retinoic acid (RA, Sigma-Aldrich). RA concentration was 0.25 µM unless stated otherwise. Spent medium was exchanged with fresh medium after 48 h.

For the RA pulse experiments (Fig. 5a) cells were first differentiated with 0.25 µM RA for the indicated amounts of time, washed three times with PBS and cultured in basal medium with 2.5 µM of the RA receptor antagonist AGN 193109 (sc-210768, Santa Cruz Biotechnology). At this concentration this antagonist completely inhibits signaling through all-trans retinoic acid[30].

For the differentiation under perturbation of various signaling pathways (Supplementary Fig. 7c) we used the MEK inhibitor PD0325901 (Stemgent, standard concentration 1 µM or dilutions thereof), GSK3 inhibitor CHIR99021 (Stemgent, standard concentration 3 µM or dilutions thereof) or mouse LIF (ESGRO, 1000 U/ml). For the experiments with MEK inhibition shown in Fig. 5b and Supplementary Fig. 7d,e we used PD0325901 at a concentration of 0.5 µM. For differentiation under inhibition of FGF signaling, shown in Supplementary Fig. 7g, h we used the FGF receptor inhibitor PD173074 (Sigma-Aldrich) at a concentration of 1 µM.

Multiple biological replicates of the differentiation of E14 cells with RA were performed, where replicates were characterized with different methods to cross-validate the results: SCRB-seq (1 replicate), SMART-seq2 (1 replicate), smFISH (3 replicates where 2 replicates used the same probe set), antibody staining (3 replicates). Morphologies similar to the ones shown in the representative images in Fig. 1a and Supplementary Fig. 5b were observed in at least 5 independent biological replicates of the experiment.

**Long-term culture of differentiated cells**. Cells that were differentiated for 96 h with RA were sorted into ectoderm-like (CD24A + /PDGFRA-) and XEN-like (PDGFRA + /CD24-) and replated on poly-D-lysine and laminin coated tissue culture dishes in basal (N2B27) medium complemented with 20 ng/ml mouse EGF (E5160, Sigma) and 10 ng/ml mouse FGF2 (SRP4038-50UG, Sigma). Ectoderm-like cells were propagated by dissociation with Accutase (Life technologies) and replating under identical conditions every 3–4 days. Floating aggregates of XEN-like cells were propagated in suspension in uncoated plastic petri dishes. Aggregates were not dissociated but the medium was refreshed typically every 4 days. Morphologies similar to the ones shown in the representative images in Supplementary Fig. 4c were observed in 3 independent biological replicates of the experiment.

**Antibody staining and FACS sorting**. We used the following antibodies: APC Rat Anti-Mouse CD24 (BD Bioscience, 562349), PE Rat Anti-Mouse CD24 (BD Bioscience, 553262), Anti-Mouse CD140a (PDGFRA) FITC (eBioscience,17-1407), Anti-Mouse CD140a (PDGFRA) APC (eBioscience,17-1401), all at a dilution of 1:1000. Cells growing in 6-well plates were washed once with PBS and then incubated in a volume of 500 µl of basal (N2B27) medium with antibodies for 30 min at 37 °C, in the dark. Subsequently, cells were washed once with PBS, 300 µl Accutase (Life Technologies) was added and cells were gently dissociated by pipetting up and down. After adding 600 µl of basal medium the cell suspension was loaded on a flow cytometer (LSR II, BD Bioscience) or cell sorter (FACSAria III, BD Bioscience). Cells growing in 10 cm dishes were first dissociated and incubated in 1 ml medium with the same incubation conditions and antibody concentrations as for adherent cells. After staining in solution, cells were spun down, the supernatant was removed and cells were resuspended in 1 ml of basal medium before flow cytometry or sorting.

Sorting gates for positive and negative populations were set by comparison to the signal measured in undifferentiated mESCs. For the experiments shown in Fig. 4b, c cells were sorted according to quartiles of CD24 signal at 48 h or terciles of CD24 signal at 72 h.

**Colony formation assay**. Cells were differentiated with or without RA as described above for various amounts of time and then replated at a density of $5 \times 10^4$ cells/well in a gelatinized 6-well tissue culture plate in 2i/L. Colonies were grown for 2 additional days, washed twice with PBS and then imaged in PBS. Remaining colonies were counted automatically by a custom made image analysis script written in MATLAB. The number of surviving colonies was normalized to the first data point (replating of untreated cells growing in 2i/L).

**Measurement of cell cycle phases**. Cells growing on gelatinized tissue culture dishes were washed twice with PBS, detached with Accutase (Life technologies) and resuspended in full medium. Formaldehyde was added to the cell suspension to a final concentration of 4%. Cells were incubated for 12 min at room temperature while being rotated and then spun down for 3 min at 90 x g. Subsequently cells were permeabilized at least over night in 70% ethanol. Cells were stained with Hoechst 33342 in PBS for 1 h and fluorescence measured on a flow cytometer (LSR II, BD Biosciences). The Dean-Jet-Fox model[59] was fit to histograms of the fluorescence signal to determine the relative lengths of the cell cycle phases reported in Supplementary Fig. 5c.

**Single cell isolation for SCRB-seq**. For each differentiation time point cells were harvested and medium removed by spinning for 5 min at 90 x g. RNA was stabilized by immediately resuspending the pelleted cells in RNAprotect Cell Reagent (Qiagen) and RNaseOUT Recombinant Ribonuclease Inhibitor (Life Technologies) at a 1:1000 dilution. Just prior to fluorescence-actived cell sorting (FACS), the cells were diluted in PBS and stained for viability using Hoechst 33342 (Life Technologies). 384-well SBS capture plates were filled with 5 µl of a 1:500 dilution of Phusion HF buffer (New England Biolabs) in water and individual cells were then sorted into each well using a FACSAria II flow cytometer (BD Biosciences) based on Hoechst DNA staining. After sorting, the plates were immediately sealed, spun down, cooled on dry ice and then stored at −80°C.

**SCRB-Seq of isolated single cells**. Frozen cells were thawed for 5 min at room temperature and cell lysis was enhanced by a treatment with proteinase K (200 µg/mL;Ambion) followed by RNA desiccation to inactivate the proteinase K and simultaneously reduce the reaction volume (50 °C for 15 min in sealed plate, then 95 °C for 10 min with seal removed).

To start, diluted ERCC RNA Spike-In Mix (1 µl of 1:$10^7$; Life Technologies) was added to each well and the template switching reverse transcription reaction was carried out using Maxima H Minus Reverse Transcriptase (Thermo Scientific), our universal adapter E5V6NEXT (1 pmol, Eurogentec):

5′-iCiGiCACACTCTTTCCCTACACGACGCrGrGrG-3′

where iC: iso-dC, iG: iso-dG, rG: RNA G, and our barcoded adapter E3V6NEXT (1 pmol, Integrated DNA Technologies):

5′-/5Biosg/ACACTCTTTCCCTACACGACGCTCTTCCGATCT[BC6] N10T30VN-3′

where 5Biosg = 5′ biotin, [BC6] = 6 bp barcode specific to each cell/well, N10 = Unique Molecular Identifiers. Following the template switching reaction, cDNA from 384 wells was pooled together, and then purified and concentrated using a single DNA Clean & Concentrator-5 column (Zymo Research). Pooled cDNAs were treated with Exonuclease I (New England Biolabs) and then amplified by single primer PCR using the Advantage 2 Polymerase Mix (Clontech) and our SINGV6 primer (10 pmol, Integrated DNA Technologies):

5′-/5Biosg/ACACTCTTTCCCTACACGACGC-3′

Full length cDNAs were purified with Agencourt AMPure XP magnetic beads (0.6x, Beckman Coulter) and quantified on the Qubit 2.0 Flurometer using the dsDNA HS Assay (Life Technologies). Full-length cDNA was then used as input to the Nextera XT library preparation kit (Illumina) according to the manufacturer's protocol, with the exception that the i5 primer was replaced by our P5NEXTPT5 primer (5 µM, Integrated DNA Technologies):

5′-AATGATACGGCGACCACCGAGATCTACACTCTTTCCCTACACGACG CTCTTCCG *A*T*C*T*-3′

where * = phosphorothioate bonds.

The resulting sequencing library was purified with Agencourt AMPure XP magnetic beads (0.6x, Beckman Coulter), size selected (300–800 bp) on a E-Gel EX Gel, 2% (Life Technologies), purified using the QIAquick Gel Extraction Kit (Qiagen) and quantified on the Qubit 2.0 Flurometer using the dsDNA HS Assay (Life Technologies). Libraries were sequenced on Illumina Hiseq paired-end flow cells with 17 cycles on the first read to decode the well barcode and UMI, a 9 cycle index read to decode the i7 Nextera barcode and finally a 46 cycle second read to sequence the cDNA.

**RNA-seq on bulk samples**. Bulk RNA-seq samples comprise complete populations at 10 time points during RA differentiation (Supplementary Fig. 1) as well as various sorted subpopulations (Fig. 4b, c) and long term cultured ectoderm- and XEN-like cells (Supplementary Fig. 4d, e). Cells were collected in RNAprotect, lysed in QIAzol (Qiagen) and total RNA was extracted and purified using Direct-zol RNA MiniPrep (Zymo Research). DGE libraries were prepared from 10 ng of extracted total RNA, using the protocol described above for SCRB-seq with the exception of using more concentrated E3V6NEXT and E5V6NEXT (10 pmol).

**SCRB-seq and bulk RNA-seq read alignment**. All second sequence reads were aligned to a reference database consisting of all mouse RefSeq mRNA sequences (obtained from the UCSC Genome Browser mm10 reference set: http://genome.ucsc.edu/), the mouse mm10 mitochondrial reference sequence and the ERCC RNA spike-in reference sequences using bwa version 0.7.4 with non-default parameter "-l 24". Read pairs for which the second read aligned to a mouse RefSeq gene were kept for further analysis if 1) the initial six bases of the first read all had quality scores of at least 10 and corresponded exactly to a designed well-barcode and 2) the next ten bases of the first read (the UMI) all had quality scores of at least 30. Digital gene expression (DGE) profiles were then generated by counting, for each microplate well and RefSeq gene, the number of unique UMIs associated with

that gene in that well. Python scripts implementing the alignment and DGE derivation are available from the authors upon request.

**SMART-seq sample preparation and read alignment.** The single-cell SMART-seq2 libraries were prepared according to the SMART-seq2 protocol[33, 60] with some modifications[61]. Briefly, total RNA from single cells sorted in lysis buffer was purified using RNA-SPRI beads. Poly(A) + mRNA from each single cell was converted to cDNA which was then amplified. cDNA was subjected to transposon-based fragmentation that used dual-indexing to barcode each fragment of each converted transcript with a combination of barcodes specific to each single cell. Barcoded cDNA fragments were then pooled prior to sequencing. Sequencing was carried out as paired-end 2 × 25 bp with 8 additional cycles for each index. Alignment of the reads and calculation of gene expression was done through the Tuxedo pipeline (Tophat, Cuffquand, Cuffnorm)[62]. Gene expression was expressed as reads per kilobase exon model per million mapped reads (RPKM).

**Computational analysis bulk RNA-seq experiments.** The bulk RNA-seq results were normalized by the total amount of reads per time point. Only those genes with non-zero mean were considered for further analysis. For k-means clustering of the temporal profiles we first determined the number of robust clusters. Stability analysis[12] indicated that there were 6 robust clusters (Supplementary Fig. 1a). We then performed gene ontology enrichment analysis using the DAVID bioinformatics resource[63] the results of which are summarized in Supplementary Fig. 1b. Only the clusters of monotonically upregulated genes (clusters 5 and 6) showed significant enrichment for GO terms related to development, morphogenesis and differentiation. The heat maps of bulk RNA-seq data depict expression relative to *Gapdh* expression (Supplementary Fig. 1c). To quantify global changes in gene expression we calculated the $L^2$ norm (Euclidean norm) for individual time points including all genes with non-zero average expression across all time points. Differences in the $L^2$ norm between time points are reported in Supplementary Fig. 1e.

To reveal the identity of sorted subpopulations (Fig. 4c) the KeyGenes algorithm[23] was used with a panel of pre-/peri-implantation tissues[21] as training set. Since there were 3 replicates per tissue in the training set, leave-one-out cross-validation had to be used instead of 10-fold cross-validation.

Expression in the long term cultured ectoderm- and XEN-like cells was compared to these tissue expression data sets from the literature: neural progenitor cells[64], neural crest cells[65], yolk sac[66] and a XEN cell line[67].

Differential expression between mESCs and ectoderm-like or XEN-like cells at 96 h (Supplementary Fig. 4e) was identified by an MA-plot based method using biological replicates for all three conditions[68].

**Computational analysis SCRB-seq experiments.** A histogram of the total number of UMIs detected per cell is shown in Supplementary Fig. 2a. To reduce the influence of technical noise we discarded cells with less than 2000 UMIs (red vertical line in Supplementary Fig. 2). This cutoff nearly minimized the upper bound of the counting error per gene (Supplementary Fig. 2b) estimated by

$$\varepsilon = \frac{1}{\sqrt{<\text{UMI}>}} \cdot \frac{1}{\sqrt{\#\text{cells}}}$$

while not significantly reducing the number of detected genes (13,720, Supplementary Fig. 2c)—defined as the number of genes, which had more than one UMI in more than one cell. Due to this cutoff 2451 out of 3456 measured cells were used for further analysis (Supplementary Fig. 2e), where these are the numbers of cells analyzed at each time point:

| Time [h] | 0 | 6 | 12 | 24 | 36 | 48 | 60 | 72 | 96 |
|---|---|---|---|---|---|---|---|---|---|
| # cells | 282 | 335 | 285 | 291 | 334 | 277 | 273 | 235 | 137 |

In individual cells with more than 2000 total UMIs 850 genes were detected on average.

For all further analyses, except the calculation of Fano factors, the data was normalized in the following way to account for differences in efficiency of transcript recovery between wells: UMI counts were divided by the total number of UMI counts per cell and then multiplied by the median of total UMI counts across all cells growing in 2i/L. For the calculation of Fano factors (Fig. 1d) UMI counts were down-sampled to 2000 UMI counts per cell. This down-sampling procedure ensured that the contribution of counting error to the Fano factors was equal for all cells from all time points. To include only those genes, which exhibited significant, biological variability, we considered the coefficient of variation (CV) of individual genes over all time points with respect to the mean expression level as well as the CVs of ERCC spike-ins with known abundance (Supplementary Fig. 2f). The increase in variability with decreased average expression reflected higher technical and counting noise for lowly expressed genes. We used the 829 genes, which had the 5% highest ratios of CV and the moving average of the CV for principal component analysis, k-means clustering and t-SNE mapping (see below).

To further characterize the performance of SCRB-seq we first compared SCRB-seq data averaged over cells for individual time points with bulk RNA-seq and found them to be strongly correlated (Supplementary Fig. 2g, Pearson correlation

ρ = 0.75). We compared 100 randomly selected pairs of cells growing in 2i/L and found that SCRB-seq measurements of individual cells were strongly correlated (Supplementary Fig. 2h, Pearson correlation ρ = 0.63). By analysis of UMI counts of ERCC spike-in RNA we determined that UMI counts scaled approximately linearly with the spiked-in transcripts—the slope of a linear fit to the log-log plot of spike ins vs. UMI counts was 0.78. The efficiency of transcript recovery as determined from the offset of that linear fit was about 0.9% (Supplementary Fig. 2i).

For principal component analysis (PCA) we considered genes, which belonged to the upregulated clusters (clusters 5 and 6, Supplementary Fig. 1a) and were among the most variable genes (Supplementary Fig. 2f). Prior to PCA expression profiles of individual genes were converted to z-scores using the average expression over all time points and the moving average of the coefficient of variation (Supplementary Fig. 2f) to preserve biological variability. PCA was performed with all cells across all time points and expression profiles of individual cells were then projected on the principal components thusly determined. The genes with the highest loadings in the first two principal components are listed in Supplementary Fig. 3a and their loadings are represented graphically in Supplementary Fig. 3b.

To discover clusters of cells we used k-means clustering including all 829 most variable genes with (1—Pearson correlation) as the distance metric. Cluster-wise assessment of stability[12] was used to determine the robustness of clusters. In particular, we calculated the Jaccard similarities between clusters found in bootstrapped samples. Clusterings resulting in Jaccard similarities close to 0.5 were considered unstable. In this way two stable clusters were found for the 96 h time point. For earlier time point cells were classified according to similarity with the clusters found at 96 h or mESCs at 0 h. In particular, we first calculated the mean expression profiles of mESCs, as well as the XEN-like and ectoderm-like subpopulations at 96 h. Then Pearson correlation was calculated between those average profiles and expression profiles of individual cells at earlier time points. A cell was classified as a particular cell type when the correlation with this particular cell type exceeded the correlation with all other cell types.

Gene expression of individual genes in the SCRB-seq data set was represented in color by normalizing to the maximum expression per time point, linear histogram stretching (1st to 99th percentile) and subsequent linear mapping to a custom colormap (Supplementary Fig. 9d).

For t-distributed stochastic neighbor embedding (t-SNE) we considered genes, which were among the most variable genes (Supplementary Fig. 2f). Prior to t-SNE mapping profiles of individual genes were converted to z-scores using the average expression over all time points and the moving average of the coefficient of variation (Supplementary Fig. 2f) to preserve biological variability. One-dimensional t-SNE maps were computed using the MATLAB Toolbox for Dimensionality Reduction (v0.8.1—March 2013) ([18], L.J.P. van der Maaten, http://lvdmaaten.github.io/drtoolbox/). Expression of *Rex1* was represented in color by normalizing to the maximum expression, linear histogram stretching (0th to 95th percentile) and subsequent linear mapping to a custom colormap.

The Fano factor reported in Fig. 1d is the Fano factor of individual genes averaged over all significantly variable genes. For this calculation down-sampled SCRB-seq data was used (see above). To determine the Fano factor of possible subpopulations, cells were first clustered by k-means clustering. Then the Fano factor was calculated separately for each cluster and averaged over all clusters. This procedure was carried out repeatedly and the resulting Fano factors were again averaged.

Hierarchical clustering of the SCRB-seq data (Supplementary Fig. 3) was performed using standard MATLAB routines. The particular clustering method was complete-linkage clustering using (1—Pearson correlation) as the distance metric.

For pseudotime ordering of cells a correlation-based pseudotime was defined by $\tau = R_{pluri} - 0.5^*(R_{ect} + R_{xen})$, where are $R_{pluri}$, $R_{ect}$ and $R_{xen}$ are the Pearson correlations of an individual expression profile with the average expression of mESCs, ectoderm-like cells at 96 h and XEN-like cells at 96 h, respectively.

For comparison with expression in in vivo tissue[21] we performed PCA on standardized in vivo data using the 829 most variable genes defined above. We then projected our standardized SCRB-seq data on the plane spanned by the first two principal components.

**Computational analysis SMART-seq2 experiments.** Only cells with at least 200000 reads per cell were used, resulting in the following numbers of cells analyzed at the respective time points:

| Time [h] | 0 | 12 | 24 | 48 |
|---|---|---|---|---|
| # cells | 82 | 86 | 89 | 82 |

For all further analyses the data was normalized in the following way to account for differences in the total number of reads between samples: RPKM for individual genes were divided by the total number of RPKM per cell and then multiplied by the median of total RPKM across all cells growing in 2i/L. Cells with high expression of *Cd24* or *Pdgfra* were classified as shown in Supplementary Fig. 9a. Out of the 82 cells measured by SMART-seq2 at 48 h, 10 were considered XEN-like

(*Pdgfra* high) and 29 ectoderm-like (*Cd24* high). To compute p-values for gene expression differences in these subpopulations we used a null model that assumes that all cells were essentially identical and gene expression differences were only due to biological and technical noise. We repeatedly sampled 10 or 29 cells, respectively, with replacement from the pool of cells which did not express *Pdgfra* or *Cd24* and calculated the average expression level for each gene. The distribution of average expression levels for each gene thusly obtained was then fit with a normal distribution. The p-value was then calculated using this normal distribution and the average expression level observed in the *Cd24* or *Pdgfra* high cells.

To account for multiple hypothesis testing we used the Benjamini-Hochberg procedure and set the false discovery rate to 0.05. Additionally, we required a minimal fold-change of 2 and an absolute expression level bigger than 1 normalized RPKM for a gene to be accepted as differentially expressed Finally, we considered only genes which were defined as transcriptional regulators by gene ontology (GO) term annotation (GO:0003700, GO:0044212, GO:0045944, GO:0006355, GO:0000981). We considered a gene to be expressed / not expressed in the pluripotent state when it was robustly expressed (normalized RPKM > 5) in at least 50% / less than 5% of the cells at 0 h.

We combined the transcriptional regulators identified in this way with pluripotency network factors[51] to arrive at a set of transcription factors which are likely relevant for the lineage decision studied here. For the calculation of co-expression (Fig. 6b, c) we considered a gene to be expressed at normalized RPKM values over 1.

**Single-molecule FISH**. Cells growing in gelatinized tissue culture dishes were washed twice with PBS, detached with Accutase (Life technologies) and resuspended in full medium. Formaldehyde was added to the cell suspension to a final concentration of 4%. Cells were incubated for 12 min at room temperature while being rotated and then spun down for 3 min at 90 x g. Subsequently cells were permeabilized at least over night in 70% ethanol. For hybridization and imaging cells were attached to chambered cover slides (Nunc Lab-Tek) coated with poly-l-lysine.

In the case of intact colonies, adherent cells were fixed for 15 min with 4% formaldehyde by adding formaldehyde to the growth medium and subsequently permeabilized in 70% ethanol.

Oligonucleotide libraries with 20-nt probes for *Nanog*, *Gbx2*, *Tbx3*, *Gata6* and *Pax6* were designed and fluorescently labeled as previously described[40]. Briefly, a home-made MATLAB script was used to design probes with close to 45% GC content. The probes were then checked for low-complexity sequences and binding to other than the desired transcript by BLAST (https://blast.ncbi.nlm.nih.gov/Blast.cgi). Suitable probes were ordered from Biosearch Technologies and fluorescently labeled using a 3'-amine modification of the oligos and amine reactive dyes (GE healthcare). The hybridization buffer used for smFISH contained 2 × SSC buffer, 25 or 40% formamide, 10% Dextran Sulphate (Sigma), E. coli tRNA (Sigma), Bovine Serum Albumin (Ambion) and Ribonucleoside Vanadyl Complex (New England biolabs). 50–75 ng of the desired probes were used per 100 μl of hybridization buffer. (The mass refers only to pooled oligonucleotides, excluding fluorophores, and is based on absorbance measurements at 260 nm). Probes were hybridized for 16–18 h at 30 °C, after which we washed cells twice for 30 min at 30 °C in wash buffer (2 × SSC, 25% formamide (for all probes except *Gbx2* and *Tbx3*) or 40% formamide (for *Gbx2* and *Tbx3*)), supplemented with Hoechst 33,342. For microscopy, we filled the hybridization chamber with a mounting solution containing 1 x PBS, 0.4% Glucose, 100 μg/ml Catalase, 37 μg/ml Glucose Oxidase, and 2 mM Trolox. Imaging was done exactly as described previously[69]. Images were taken on a NIKON Ti-E inverted fluorescence microscope equipped with a Roper scientific PIXIS 1024B camera and a 100x oil immersion objective (numerical aperture 1.49). Custom filters (Omega Optical) were used for imaging TMR and Alexa 594 and a standard filter (Chroma) for Cy5. Exposure times ranges between 1 and 3 s and the distance between planes in a z-stack was 0.3 μm Home-made MATLAB scripts were used for image analysis. Cells positive for one of the assayed genes were classified as shown in Supplementary Fig. 10b.

**Quantification of the flow cytometry experiments**. The distribution of cells in the space of CD24 and PDGFRA expression was modeled by the sum of 4 bivariate normal distributions. This model has in principle 19 free parameters (8 for the means, 8 for the standard deviations and 3 for the size of the relative contributions). To ensure robust fitting to the date we reduced the number of parameters to 9 by keeping the standard deviations constant and only allowing 4 different values for the means.

$$p\left(x, y; f_1, f_2, f_3, \mu_x^{low}, \mu_y^{low}, \mu_x^{high}, \mu_y^{high}, \sigma_x, \sigma_y\right)$$
$$= f_1 \cdot N\left(x, y; \mu_x^{low}, \mu_y^{low}, \sigma_x, \sigma_y\right) + f_2 \cdot N\left(x, y; \mu_x^{low}, \mu_y^{high}, \sigma_x, \sigma_y\right)$$
$$+ f_3 \cdot N\left(x, y; \mu_x^{high}, \mu_y^{low}, \sigma_x, \sigma_y\right) + \left(1 - \sum_{i=1}^{3} f_i\right) \cdot N\left(x, y; \mu_x^{high}, \mu_y^{high}, \sigma_x, \sigma_y\right)$$

$N(x, y, \mu_x, \mu_y, \sigma_x, \sigma_y)$ is a bivariate normal distribution in x and y (PDGFRA and CD24 expression, respectively) with mean $(\mu_x, \mu_y)$ and standard deviation $(\sigma_x, \sigma_y)$. This model was fit to a reference data set (typically untreated control cells after 96 h of RA exposure) by maximizing the log-likelihood $-\log(p)$. To subsequently calculate the size of the fractions $f_i$ for a particular sample we first calculated the probabilities that the expression values $(x, y)$ found in a particular cell were drawn from one the 4 normal distributions $N(x, y, \mu_x, \mu_y, \sigma_x, \sigma_y)$. The cell was then ascribed to the distribution from which it was most likely drawn.

**Stochastic simulation of the lineage transition**. We simulated the differentiation process using a discretized version of the Langevin equation describing the system (Euler method):

$$dX = \left(a_X \frac{X^n}{\theta^n + X^n} + b\frac{\theta^n}{\theta^n + E^n} - kX\right)\Delta + \sqrt{D\Delta}\,\mathcal{N}(0, 1)$$
$$dE = \left(a_E \frac{E^n}{\theta^n + E^n} + b\frac{\theta^n}{\theta^n + X^n} - kE\right)\Delta + \sqrt{D\Delta}\,\mathcal{N}(0, 1)$$

$X$ and $E$ indicate the expression levels of the XEN and ectoderm programs respectively. $N(0, 1)$ indicates a Wiener process with mean 0 and standard deviation 1. D sets the strength of gene expression noise and $\Delta$ determines the size of the time step. After initializing $X$ and $E$ randomly between 0 and 0.1 we first equilibrated the system for 100 iterations. Subsequently, we propagated the system for 200 additional iterations. To relate the simulation to experimental time scales, the end point of the simulation was taken to be at 96 h. To model the exit from pluripotency the degradation parameter k was switched from a high value ($k = 10$) to a low value ($k = 1$) after 12 h (25 iterations), which allowed X and E to increase. To model timed application of RA the auto-activation parameter for the XEN program $a_X$ was switched at various points in time (no RA: $a_X = 0$; RA: $a_X = 0.5$). For each condition we generated 10,000 trajectories and counted the number of trajectories that ended at the XEN or ectoderm attractor (see Supplementary Fig. 8b). The relative frequency of trajectories ending at the XEN attractor is reported in Fig. 5e.

**Used parameters**.
$n = 4$
$\theta = 0.5$
$\Delta = 0.05$

pluripotency: $k = 10$
differentiation: $k = 1$

$a_E = 0.5$
no RA:, $a_X = 0$
RA: $a_X = 0.5$

In Supplementary Fig. 8b
low noise: D = 0.0001
high noise: D = 0.01

**Code availability**. The MATLAB scripts used for data analysis and simulations are freely available on request from the corresponding author.

**Data availability**. All raw and processed data is freely available from the GEO repository (https://www.ncbi.nlm.nih.gov/geo/) under accession number GSE79578.

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

## Acknowledgements
S.S. was supported by the Netherlands Organisation for Scientific Research (NWO/OCW), as part of the Frontiers of Nanoscience program and by an NWO Rubicon award. J.G. was supported by the Boehringer Ingelheim Foundation as well as a Jerome and Florence Brill Graduate Student Fellowship. R.J. was supported by NIH grants HD 045022 and RO1-CA084198. A.v.O. was supported by an ERC Advanced grant (ERC-AdG 294325-GeneNoiseControl) and an NWO Vici award.

## Author contributions
S.S. designed and performed experiments, analyzed data and wrote the manuscript, J.G. performed experiments and edited the manuscript, M. S. performed experiments, T.S.M. supervised the project, R.J. and A.v.O. supervised the project and edited the manuscript.

## Additional information

**Competing interests:** The authors declare no competing financial interests.

