## [Peer Review file · Nature Communications]

Reviewers' comments:

Reviewer #1 (Remarks to the Author):

In this study the authors apply single cell RNA-seq technology to analyse the entry into differentiation of pluripotent mouse embryonic stem (ES) cells in adherent culture. This is in principle a timely study because previous analyses have mostly been at the population level and single cell studies have either been lower throughput (Singer et al., 2014) or used serum-based cultures which are heterogeneous (Kolodziejczyk et al., 2015). The authors find that two different cell populations are generated after 96hrs of treatment with retinoic acid. They go on to define a time window of responsiveness to retinoic acid and suggest a transitory period exists during which cell specification may be altered. They then infer from the single cell time course data that there is a lineage bifurcation between 24-36hrs and propose a model of initial stochastic gene expression of "early lineage" genes creating lineage bias that is subsequently consolidated into commitment by expression of "late lineage" genes. These propositions are more or less interesting and generally not unreasonable. Unfortunately, however, they are not supported by examination of the data and furthermore the biological relevance of the experimental system is questionable.

1) The use of retinoic acid is justified as accelerating differentiation. Actually this effect is modest (Supp Fig 1f). More to the point is the fact that retinoic acid is a pleiotropic effector which can induce multiple differentiation events including atypical differentiation of ES cells into extraembryonic endoderm-like cells. This occurs through an unknown pathway that appears unrelated to events in the embryo. The authors discuss at length the bifurcation between XEN and ectoderm lineages. Even if this occurs in their system, (and I do not see how the authors exclude independent trajectories), it would be artificial because pluripotent cells in the embryo are not confronted with a fate decision between extraembryonic endoderm, which form before implantation, and ectoderm, which forms some days later after implantation.

2) The authors explicitly designate the 12h time point after 2i withdrawal/RA addition as the exit from pluripotency. This is factually incorrect, because at 12h all cells are clonogenic when plated back to 2i. In fact the data in Supp Fig 1 show increased self-renewal at the 12hr time point, which is in itself curious. The data show that loss of self-renewal ("exit from pluripotency occurs sometime between 12h-48h across the population and seems to be asynchronous at the single cell level"). Furthermore, the gene expression classification in Fig 2b indicates that at 36hrs more than 80% of cells are "ESC-like".

3) The conclusion that there are only 2 cell types at 96h time point is not fully convincing. In the PCA plot (Fig 2a), 96h cells do not present as 2 clusters, rather as a continuum. The authors say they have performed k means clustering and stability analysis. However, a heatmap of gene expression with a cluster dendrogram should be provided for 96h cells and for selected earlier time points. This would allow the biologist to evaluate the degree of relationships between single cells, how many subpopulations exist and their identities based on gene expression profiles. Is detection of only 850 expressed genes per cell really sufficient to assign identities. This point is crucial to clarify, as the paper is built on the initial classification of cell types. Without these analyses, the rest of the data is hard to interpret.

4) Following on from the previous point, the classification of cell type 1 as "ectoderm-like" is misleading, as primary markers used to define these cells (prtg, mdk, cd24) are not specific to ectoderm. And what do the authors actually mean by ectoderm (neuroectoderm?). Moreover, PC2 genes that are highly expressed in so-called "ectoderm-like cells" include Smoc2 (a Sparc matrix protein) and Foxa2, both expressed in the endoderm (Supp Fig 3c). Thus, it is likely that the "ectoderm-like" population contains some cells that are transiting either to become XEN-like cells or possibly definitive endoderm. Furthermore, Cyclin D2 (ccnd2) is the second most important contributor to PC2, suggesting "ectoderm-like cells" may be in a highly proliferative state, compared to XEN cells. If this is indeed the case, interpretation of the rest of the data would be different. A more detailed gene expression characterisation of the cell populations at each time point is crucial.

5) Tbx3 and Gbx2 are identified as early lineage specifiers for XEN and ectoderm fates and attention drawn to their expression in undifferentiated ES cells. It is very surprising that the

authors do not consider the findings that both Tbx3 and Gbx2 are functional players in ES cell self-renewal (Dunn et al., 2014; Niwa et al., 2009; Tai and Ying, 2013).

6) Paragraph 3 of the Results section states: "Interestingly, we found biases in the differentiation outcome depending on the timing of exit from pluripotency. Cells that down-regulated the mESC expression profile early (before 36 h) were biased towards the ectoderm lineage, while cells that exited the pluripotent state late (after 48h) adopted a XEN-like transcriptional program. This observation indicated that commitment to the two lineages did not occur simultaneously and that the lineage decision was initially biased towards ectoderm."

I can find no data to support this argument. As discussed in point 2 exit from pluripotency starts sometime between 12-24h and continues until 48h across the population. Since the cell behave asynchronously, it is only possible to know when a cell exits pluripotency within this time window and its subsequent fate by single cell tracking. The biological meaning of such a conclusion is also obscure because in the embryo embryonic endoderm segregates long before ectoderm.

7) Relating to the asynchronous nature of ES cell entry into differentiation, it seems surprising that the authors did not at least investigate pseudotime ordering.

Dunn, S. J., Martello, G., Yordanov, B., Emmott, S. and Smith, A. G. (2014). Defining an essential transcription factor program for naïve pluripotency. *Science* 344, 1156-1160.

Kolodziejczyk, A. A., Kim, J. K., Tsang, J. C., Ilicic, T., Henriksson, J., Natarajan, K. N., Tuck, A. C., Gao, X., Buhler, M., Liu, P. et al. (2015). Single Cell RNA-Sequencing of Pluripotent States Unlocks Modular Transcriptional Variation. *Cell Stem Cell* 17, 471-85.

Niwa, H., Ogawa, K., Shimosato, D. and Adachi, K. (2009). A parallel circuit of LIF signalling pathways maintains pluripotency of mouse ES cells. *Nature* 460, 118-22.

Singer, Zakary S., Yong, J., Tischler, J., Hackett, Jamie A., Altinok, A., Surani, M. A., Cai, L. and Elowitz, Michael B. (2014). Dynamic Heterogeneity and DNA Methylation in Embryonic Stem Cells. *Molecular Cell* 55, 319-331.

Tai, C. I. and Ying, Q. L. (2013). Gbx2, a LIF/Stat3 target, promotes reprogramming to and retention of the pluripotent ground state. *J Cell Sci*.

Reviewer #2 (Remarks to the Author):

The report from Semrau et al. describes the first stages of embryonic stem cell differentiation in the presence of retinoic acid. Single cell expression profiling revealed that exit from pluripotency is accompanied by a lineage bifurcation and a transient susceptibility to lineage specifying signals. The authors found that there is a sharp increase in gene expression variability during the initial phase of lineage specification. This paper also explored the role of retinoic acid on XEN cell specification and reported two classes of TFs with potential roles: early-expressed TFs for lineage biasing and late expressed TFs for lineage commitment. The strength of the report is in a careful and detailed single-cell level study to characterize exit from pluripotency and lineage commitment during ESC differentiation. The weakness is that none of the inferred mechanisms and conclusions reached in the manuscript are experimentally validated.

1. Most conclusions of this paper are based on single-cell RNA-seq. The authors also performed bulk RNA-seq (10 time points) and the same analysis (k-means clustering and stability analysis, Supplementary Figs. 1 and 3a,b) side by side with single-cell RNA-seq. However, the results from bulk RNA-seq were not much presented except the first paragraph in the Results section (page 2). The authors should directly compare and discuss the single-cell RNA-seq data and the bulk RNA-seq results. Specifically, in page 2-3, the authors concluded that "Quantification of single-cell

variability of gene expression confirmed a steep increase in variability between 12 h and 24 h (Fig. 1c)". However, bulk RNA-seq analysis also shows similar results of a steep increase of gene expression changes between 12 h and 24 h (ectoderm genes, red line in Supplementary Fig. 1b). Ectoderm-like and XEN-like cell types in single-cell RNA-seq (Fig. 2b) seem to be similar as cluster 5 and 6 in bulk RNA-seq (Supplementary Fig. 3a), respectively. The authors should present how single-cell analysis is similar or different compared to bulk RNA-seq analysis.

2. From single-cell RNA-seq expression data at 96 h in Fig. 1b, the authors identified the "two robust clusters" as two distinct cell types (ectoderm-like and XEN-like) based on k-means clustering and stability analysis. Most of the follow-up analyses (e.g. principal component analysis, gene regulatory network modeling, etc.) in this paper are based on this classification of two groups of genes. On the other hand, from bulk RNA-seq analysis (Supplementary Figs. 3a,b) and stability analysis (page 12 in Supplementary Methods), the authors found "6 robust clusters" at 96 h. This is somewhat surprising and the authors should explain in detail how they arrived to the conclusion that there are only "two robust clusters" at 96 h based on their single-cell RNA-seq data. For example, it is conceivable to subdivide Cell type 2 (ectoderm-like) into two groups of genes (Fig. 1b) as well as cell type 1 (XEN-like) (Fig. 2a).

3. To assess susceptibility to lineage specifying signals (Fig. 3), the authors modulated signals during differentiation and quantified cell type frequencies using ectoderm and XEN marker, *cd24* and *pdgfra*, respectively. However, *pdgfra* is not identified in the top25 genes of the expression profile of principal component 1 in this paper (XEM-like; Supplementary Fig. 3c). The threshold for these top 25 XEM-like genes is quite low (loadings 0.06) compared to top 25 ectoderm-like genes (loadings 0.11), suggesting that *pdgfra* may not be a good XEN marker to quantify cell type frequency in this paper. The authors may need to use different markers for XEN-like cell quantification.

4. The authors observed that continuous treatment with RA promotes appearance of XEN cells. They also identify groups of transcription factors that might be involved in the cell identity specification. The obvious question is whether RA treatment influences expression of any of these putative "patterning" factors. Additional question is that differentiation without RA seems to be much more homogeneous towards neuroectodermal lineage. It would be interesting to examine whether these cells do not undergo similar stage of "Expression noise" as cells exposed to RA.

5. For these transcription factor analysis (Fig. 4), Supplementary Methods describe that the authors used only 39 out of 82 cells (page 15 in Supplementary Methods, "Out of the 82 cells measured by SMART-seq at 48 h, 10 were considered XEN-like and 29 ectoderm-like"). The authors may find other groups of transcription factors from the rest of cells (43 out of 82 cells), or at least need to explain what expression patterns for these 43 of the filtered cells look like.

6. Substantial amount of results in the paper are shown in the Supplementary Figures. However, Some of Supplementary Figures in this paper are not ordered properly, not matched with figure contents, or nothing mentioned in the paper although some figures present. The following are some of examples. The authors should arrange and explain carefully Supplementary Figures according to the results in the this paper.

a. In page 4 in the text, Supplementary Figs. 3a and 3b should show that "Cell type frequencies were quantified after 96 h using antibody staining". However, Supplementary Figs. 3a and 3b do not show these results (Antibody staining in Supplementary Figs. 3a and 3b is missing).

b. "Bulk RNA-seq" clustering analyses shown in Supplementary Figs. 3a and 3b is not mentioned in the paper.

c. Supplementary figs. 2a-e and 2g-i are not mentioned in the main text or figure legends.

d. Supplementary Figs. 4a and 4b are not mentioned in the main text, figure legends, or Supplementary Methods.

e. Supplementary Figs. 8a and 8b related to Fig. 4 (page 23) are missing and not shown anywhere in the paper.

f. In the figure legend for Supplementary Fig. 3, "d, Graphical representation of the data in e". However, "the data in e" is unclear. Does e indicate Fig. 2e or 3e or missing?

Reviewer #3 (Remarks to the Author):

In this report, Semrau and colleagues present a detailed analysis of transcriptional changes during the exit of pluripotency and cell fate commitments in mouse embryonic stem cells. The experiments are well designed, carefully executed and nicely described and discussed. Unusual for a work with transcriptional analysis at the level of single cells, there is a biological narrative. There is little question that this work provides significant observations and a good model and therefore I am happy to support publication. However, there are a number of issues that I would like the authors to consider.

I suspect that many pluripotency student will be very surprised to see XEN appear as an alternative fate in the exit of pluripotency. Most of the works on this subject consider an alternative between ectoderm (or neuroectoderm) and mesendoderm and therefore it will surprise them to see the fate choice revealed by the analysis. The authors should comment on this and make a reference to the plausibility of the observation. From the perspective of this reviewer, the observation is sound. The time course of the differentiation, the quality of the data, the consistency of the results, all point to the fact that this is a genuine observation. Why then is there no trace of this fate decision in the literature? It will be interesting and significant that the authors discussed this matter. I would suggest that they notice that they start from cells grown in 2i+LIF and that it has been suggested that long term exposure to this medium can turn ES cells into quasi-totipotent cells (Morgani et al. Totipotent embryonic stem cells arise in ground-state culture conditions. *Cell Rep.* 2013 Jun 27;3(6):1945-57). Could it therefore be that, in their experiments what they are seeing is a fate choice from an ICM like population in which cells choose between a XEN fate and an advanced -RA induced- epiblast fate? It would be really helpful to readers and, also to the authors, to comment on the unusual nature of this observation and some possible explanations. In this regard, there is clearly a hint of neural fate in the XEN population but, in the face of what is shown, it is not that clearly neural; could it be a modified epiblast? Have they looked at epiblast markers such as *Fgf5*, *Otx2*, *Pou3f1*, 2? Can they rule out that they are not seeing an EPI v XEN decisions?

In the argument raised above an important issue is that the experiments are started from 2i+LIF, thus something that would be of interest is to see what happens if the same experiment was started from LIF + BMP or Serum + LIF. I am raising this as a thought experiment and not as something the authors should do, though they may have some relevant results.

These matters aside, there are a number of small issues:

The authors use Martello et al (ref 30) as their reference for pluripotency network. There are others and there is no reason, to quote this exclusively.

In page 4, the discussion of the experiments on the effect of MEKi on RA induction could and should be explained better.

In page 5, the authors talk about a phase in which cells coexpress XEN and neuroectoderm markers; it would be helpful if they could provide some examples of this in the main text. This is an important point and should be illustrated properly.

In the discussion, the authors mention the possibility that some pluripotency genes might be involved in differentiation and make two references 31 and 32. They might want to also include Malleshaiah et al. *Nac1 Coordinates a Sub-network of Pluripotency Factors to Regulate Embryonic Stem Cell Differentiation.* *Cell Rep.* 2016 Feb 9;14(5):1181-94, which shows some relevant data

and present some ideas and models related to those under discussion here.

In Supplementary Fig 3: d it says: says d, Graphical representation of the data in e. It probably means 'in c' rather than 'in e'.

In Supplementary Fig6B, some of the proteins listed as TFs are NOT TFs e.g axin2, fgfr2, tsc22d4, dedd2, fgf10, maybe there are more. The authors should make sure that if they are talking about TFs what they refer to are TFs.

Please find below our detailed responses to the reviewers' concerns. In the manuscript all sections that are substantially new are highlighted in blue.

Reviewer #1 (Remarks to the Author):

1) The use of retinoic acid is justified as accelerating differentiation. Actually this effect is modest (Supp Fig 1f). More to the point is the fact that retinoic acid is a pleiotropic effector which can induce multiple differentiation events including atypical differentiation of ES cells into extraembryonic endoderm-like cells. This occurs through an unknown pathway that appears unrelated to events in the embryo.

The reviewer is correct in pointing out that RA has a pleiotropic effect and according to current knowledge RA is not involved in the epiblast/primitive endoderm decision *in vivo*. Nevertheless, RA is widely used in *in vitro* assays since it is a powerful inducer of differentiation (Gudas et al., J. Cell. Phys., 2011). Furthermore, specification of XEN cells has been observed also in differentiation by LIF withdrawal (Morgani et al., Cell Rep., 2013, Klein et al., Cell, 2015). Hence, it is not an idiosyncratic artefact of RA exposure.

In the revised version of the manuscript we show that RA driven differentiation involves the same signaling pathways (FGF, MAPK/Erk) that are relevant in the embryo. In particular, we have carried out new differentiation experiments under FGF receptor inhibition that show the importance of this pathway and a clear difference from MEK inhibition (Supplementary Fig. 7c-h). As we now point out in the Discussion section, our experimental system recapitulates *in vivo* measurements of the susceptibility to signaling inputs over time (Saiz et al., Nature Comm., 2016). Finally, in the revised version of the manuscript we report new experiments using serum and LIF conditions before differentiation. These experiments show that our experimental system is sensitive to changes in culture conditions that modify the developmental potential of mESCs (Supplementary Figure 7a-b). RA driven differentiation is thus a useful system to explore the developmental potential of different pluripotent states.

The authors discuss at length the bifurcation between XEN and ectoderm lineages. Even if this occurs in their system, (and I do not see how the authors exclude independent trajectories), it would be artificial because pluripotent cells in the embryo are not confronted with a fate decision between extraembryonic endoderm, which form before implantation, and ectoderm, which forms some days later after implantation.

We are thankful for this remark and a similar remark of reviewer 3, as they prompted us

to re-examine our single-cell RNA-seq data set. Indeed, we found an initial bifurcation between epiblast-like and XEN-like cells. In particular, pseudotemporal ordering of our single-cell RNA-seq data set (Supplementary Fig. 6) revealed the transient expression of early epiblast-markers which continued somewhat in the ectoderm-branch. We confirmed this new finding using 12 new bulk RNA-seq data sets (Fig. 2d-e). Specifically, we sorted cells into subpopulations according to CD24 and PDGFRA expression and profiled them by bulk RNA-seq. We found that PDGFRA negative/CD24 high cells expressed early epiblast markers at 48 h and 72 h but not anymore at 96 h. With the KeyGenes algorithm (Roost et al., Stem Cell Rep. 2016) we identified the CD24 high cells as E5.5 epiblast. In conclusion, cells indeed bifurcate between epiblast and XEN initially. However, due to continuous exposure to RA, epiblast-like cells then do quickly differentiate further to neuroectoderm. In the revised manuscript we further support this finding by two new bulk RNA-seq data sets. In particular, cells were purified as ectoderm- or XEN like at 96 h, cultured further separately and then profiled by bulk RNA-seq (Supplementary Fig. 4d-f). We found that these two cell lines expressed markers for neuroectoderm/neural crest or primitive endoderm, respectively, and had expression profiles similar to corresponding tissues *in vivo*.

2) The authors explicitly designate the 12h time point after 2i withdrawal/RA addition as the exit from pluripotency. This is factually incorrect, because at 12h all cells are clonogenic when plated back to 2i. In fact the data in Supp Fig 1 show increased self-renewal at the 12hr time point, which is in itself curious. The data show that loss of self-renewal (“exit from pluripotency occurs sometime between 12h-48h across the population and seems to be asynchronous at the single cell level”). Furthermore, the gene expression classification in Fig 2b indicates that at 36hrs more than 80% of cells are “ESC-like”.

We meant to indicate that the FIRST cells exit pluripotency after 12 h. Thanks to the reviewer’s comment we realized that we did not express this idea clearly. As the exit from pluripotency is indeed asynchronous, we now indicate the period from 24 h to 48 h as the exit from pluripotency (Fig. 1c). The reviewer furthermore points out correctly that the decrease in cells classified as mESC-like by their expression profile (Fig. 2c) lags behind the loss of self-renewal capacity in 2i (Fig. 5d). These two results are consistent, though, since the down-regulation of a few critical genes early in differentiation is likely enough to compromise self-renewal capacity, while, overall, the expression profile is still similar to mESCs in 2i.

3) The conclusion that there are only 2 cell types at 96h time point is not fully convincing. In the PCA plot (Fig 2a), 96h cells do not present as 2 clusters, rather as a continuum. The authors say they have performed k means clustering and stability analysis. However, a heatmap of gene expression with a cluster dendrogram should be provided for 96h cells and for selected earlier time points. This would allow the biologist to evaluate the degree of relationships between

single cells, how many subpopulations exist and their identities based on gene expression profiles.

As requested by the reviewer we have created cluster dendrograms (Supplementary Fig. 3c), which support our original statement. At 96 h the two highest clusters in the hierarchical clustering largely overlap with the clusters we found by k-means clustering. In one cluster we find higher expression of gene cluster 6 (which contains XEN markers), in the other cluster genes of gene cluster 5 (which contains ectoderm markers) are expressed more strongly. Correspondingly, genes with the highest contributions to principal components 1 or 2 respectively (Fig. 1b), show different expression levels in the two clusters. Hence, hierarchical clustering confirms the existence of two cell types at 96 h. At early time points no clear clusters of significant size are visible. We also carried out k-means clustering with 2,3 and 4 clusters for all time points and calculated the expression variability at all time points (Fig. 1d). This analysis shows that assuming 3 clusters does not result in a large reduction of variability compared to 2 clusters. Our method can thus robustly identify 2 clusters.

Is detection of only 850 expressed genes per cell really sufficient to assign identities. This point is crucial to clarify, as the paper is built on the initial classification of cell types. Without these analyses, the rest of the data is hard to interpret.

To relieve the reviewer's concerns we created additional bulk RNA-seq data sets of purified ectoderm- and XEN-like cells at 96 h and after extended culture of these purified cell types (as already described above), see Supplementary Figure 4. In this bulk RNA-seq data set we detect on average 13000 genes in each sample. We believe that these new data sets clearly support our interpretation of the cell clusters at 96 h as (neuro)ectoderm-like and XEN-like. In particular, we found that ectoderm-like cells expressed markers for neural progenitors and neural crest and had an overall expression profile that was similar to neural progenitors and neural crest cells *in vivo*. XEN-like cells strongly expressed primitive endoderm markers and their expression profile was similar to an embryo-derived XEN cell line and yolk sac tissue *in vivo*.

4) Following on from the previous point, the classification of cell type 1 as “ectoderm-like” is misleading, as primary markers used to define these cells (prtg, mdk, cd24) are not specific to ectoderm. And what do the authors actually mean by ectoderm (neuroectoderm?). Moreover, PC2 genes that are highly expressed in so-called “ectoderm-like cells” include Smoc2 (a Sparc matrix protein) and Foxa2, both expressed in the endoderm (Supp Fig 3c). Thus, it is likely that the “ectoderm-like” population contains some cells that are transiting either to become XEN-like cells or possibly definitive endoderm. Furthermore, Cyclin D2 (ccnd2) is the second most important contributor to PC2, suggesting “ectoderm-like cells” may be in a highly proliferative state, compared to XEN

cells. If this is indeed the case, interpretation of the rest of the data would be different. A more detailed gene expression characterisation of the cell populations at each time point is crucial.

Concerning this point, it is important to note that the principal components were calculated using the data from all time points (as we now indicate clearly in the captions and in the Methods). Therefore, it is not surprising to find genes which are not representative of the final neuroectoderm lineage but maybe the transient epiblast-like state. We chose to calculate the principal components across the whole data set since that allowed us to show the gene expression dynamics across the whole time course in the same principal component space (Fig. 2a). Hence, the principal components might reflect transiently expressed genes as well as genes that are up-regulated in the final, differentiated cell types. Secondly, we believe that it is difficult to draw final conclusions about the identity of cell types from a few genes. *Foxa2* is not only expressed in the endoderm but also in the floor plate of the developing neural tube (Placzek et al., Nature rev. Neuroscience, 2005); *Smoc2* is broadly expressed in many different tissues, including the brain (Vannahme, Biochem. J., 2003) and Cyclin D2 is expressed in ectoderm cells in the vicinity of the primitive streak and later in the mid-neural tube (Wianny et al., Dev. Dyn., 1998). As already stated in our response to the previous comment, we have now complemented our single-cell RNA-seq measurements with more comprehensive bulk RNA-seq measurements of purified subpopulations, which confirmed our initial cell type assignment.

5) *Tbx3* and *Gbx2* are identified as early lineage specifiers for XEN and ectoderm fates and attention drawn to their expression in undifferentiated ES cells. It is very surprising that the authors do not consider the findings that both *Tbx3* and *Gbx2* are functional players in ES cell self-renewal (Dunn et al., 2014; Niwa et al., 2009; Tai and Ying, 2013).

We thank the reviewer for pointing us to these references. We now mention the role of *Gbx2* and *Tbx3* in pluripotency.

6) Paragraph 3 of the Results section states: “Interestingly, we found biases in the differentiation outcome depending on the timing of exit from pluripotency. Cells that down-regulated the mESC expression profile early (before 36 h) were biased towards the ectoderm lineage, while cells that exited the pluripotent state late (after 48h) adopted a XEN-like transcriptional program. This observation indicated that commitment to the two lineages did not occur simultaneously and that the lineage decision was initially biased towards ectoderm.”

I can find no data to support this argument. As discussed in point 2 exit from pluripotency starts sometime between 12-24h and continues until 48h across the population. Since the cell behave asynchronously, it is only possible to know

when a cell exits pluripotency within this time window and its subsequent fate by single cell tracking. The biological meaning of such a conclusion is also obscure because in the embryo extremes embryonic endoderm segregates long before ectoderm.

We do agree with the reviewer that live cell tracking would be needed to convincingly support this point. Hence, we have removed the statement in question from the manuscript.

7) Relating to the asynchronous nature of ES cell entry into differentiation, it seems surprising that the authors did not at least investigate pseudotime ordering.

We are indebted to the reviewer for this suggestion. The pseudotemporal ordering (Supplementary Fig. 6) turned out to be a great way to reveal the transient epiblast-like gene expression pattern. In particular, we found that prior to the occurrence of ectoderm-like and XEN-like cells in pseudotime, several pluripotency factors were already down-regulated, while simultaneously the expression of early epiblast markers (like *Pou3f1*, *Dnmt3a*, *Zic2*) increased. The expression of some of these markers persisted transiently in the ectoderm-branch of the bifurcation. Expression of neuroectodermal markers (like *Ascl1* or *Nes*) occurred later in pseudotime in the ectoderm branch. The expression of primitive endoderm markers was restricted to the XEN branch of the bifurcation and it increased with increasing pseudotime. Hence, the pseudotime analysis indicated that cells initially decided between epiblast and XEN.

Reviewer #2 (Remarks to the Author):

1. Most conclusions of this paper are based on single-cell RNA-seq. The authors also performed bulk RNA-seq (10 time points) and the same analysis (k-means clustering and stability analysis, Supplementary Figs. 1 and 3a,b) side by side with single-cell RNA-seq. However, the results from bulk RNA-seq were not much presented except the first paragraph in the Results section (page 2). The authors should directly compare and discuss the single-cell RNA-seq data and the bulk RNA-seq results. Specifically, in page 2-3, the authors concluded that “Quantification of single-cell variability of gene expression confirmed a steep increase in variability between 12 h and 24 h (Fig. 1c)”. However, bulk RNA-seq analysis also shows similar results of a steep increase of gene expression changes between 12 h and 24 h (ectoderm genes, red line in Supplementary Fig. 1b).

In the revised version of the manuscript we have substantially extended the discussion of the bulk RNA-seq data set, especially at the beginning of the Results section. We also included a quantification of global gene expression changes at the population level

(Supplementary Fig. 1e). This analysis shows a wave of changes between 24 h and 36 h which provides an additional indicator for the exit from pluripotency. We have further facilitated the comparison between bulk and single-cell RNA-seq by grouping and highlighting genes by the gene clusters derived from bulk RNA-seq (Supplementary Figs. 3c and 6d). We believe that this ordering helps to reveal the role of gene clusters 5 and 6 (see our reply to the next comment).

Ectoderm-like and XEN-like cell types in single-cell RNA-seq (Fig. 2b) seem to be similar as cluster 5 and 6 in bulk RNA-seq (Supplementary Fig. 3a), respectively. The authors should present how single-cell analysis is similar or different compared to bulk RNA-seq analysis.

We thank the reviewer for this observation. The new supplementary figures 3c and 6d now visualize the suggested relationship and further support the cell type identification. Genes in gene cluster 5, which contains ectoderm markers, are more highly expressed in cells that we classified as ectoderm-like. Genes in gene cluster 6, which contains XEN markers, are preferentially expressed in XEN-like cells. This association between gene clusters and cell clusters is evident for hierarchical clustering and k-means clustering of cells (Supplementary Figure 3c), as well as supervised classification of cells (Supplementary Figure 6). The clustering of genes according to their temporal expression profiles measured by bulk RNA-seq thus helped to support the identification of cell types in the single-cell RNA-seq data set.

2. From single-cell RNA-seq expression data at 96 h in Fig. 1b, the authors identified the “two robust clusters” as two distinct cell types (ectoderm-like and XEN-like) based on k-means clustering and stability analysis. Most of the follow-up analyses (e.g. principal component analysis, gene regulatory network modeling, etc.) in this paper are based on this classification of two groups of genes. On the other hand, from bulk RNA-seq analysis (Supplementary Figs. 3a,b) and stability analysis (page 12 in Supplementary Methods), the authors found “6 robust clusters” at 96 h. This is somewhat surprising and the authors should explain in detail how they arrived to the conclusion that there are only “two robust clusters” at 96 h based on their single-cell RNA-seq data. For example, it is conceivable to subdivide Cell type 2 (ectoderm-like) into two groups of genes (Fig. 1b) as well as cell type 1 (XEN-like) (Fig. 2a).

Due to the reviewer’s remark we realized that we did not discriminate carefully enough between “gene clusters” and “cell clusters” in the text and improved the manuscript in that respect. The “6 robust clusters” are clusters of genes that are based on the temporal expression profiles measured by bulk RNA-seq. We now state that explicitly in the manuscript. The “two robust clusters” are clusters of cells at 96 h. The existence of

gene clusters does not imply an equal number of cell clusters. For example, in a (hypothetical) population of perfectly identical cells, one might find several clusters of genes, where the genes in each cluster have similar dynamics. These dynamics would be the same in all (perfectly identical) cells and hence there would be just one cell cluster (which comprises the whole population). Additionally, the first 4 of the gene clusters observed here are strongly down-regulated by the 96 h time point. Clusters 5 and 6, on the other hand, increase towards the end and these are the clusters that are expressed in either ectoderm-like and XEN-like cells respectively. Thus, in our case, gene clusters 5 and 6 and the two cell clusters correspond.

To further support our result that there were 2 clusters of cells we performed k-means clustering with 2,3 and 4 clusters at all time points and calculated the expression variability at all time points (Fig. 1d). This analysis showed that assuming 3 clusters does not result in a large reduction of variability compared to 2 clusters. Hence, while our method is able to resolve 2 clusters, we cannot rule out that there are sub-clusters, which a more precise method might reveal.

3. To access susceptibility to lineage specifying signals (Fig. 3), the authors modulated signals during differentiation and quantified cell type frequencies using ectoderm and XEN marker, *cd24* and *pdgfra*, respectively. However, *pdgfra* is not identified in the top25 genes of the expression profile of principal component 1 in this paper (XEM-like; Supplementary Fig. 3c). The threshold for these top 25 XEM-like genes is quite low (loadings 0.06) compared to top 25 ectoderm-like genes (loadings 0.11), suggesting that *pdgfra* may not be a good XEN marker to quantify cell type frequency in this paper. The authors may need to use different markers for XEN-like cell quantification.

Indeed, *Pdgfra* does not appear in the top 25 genes in PC1, probably due to its low expression level. Since we wanted to be able to sort live cells we needed a surface marker and *Pdgfra* is the earliest known marker of the primitive endoderm lineage *in vivo* (Artus et al., Development, 2010). Our new bulk RNA-seq measurements of cells sorted by PDGFRA expression (Fig. 2d-e, Supplementary Fig. 4d-e) clearly validate our choice. PDGFRA+/CD24- cells at 72 h and 96 h are identified as E4.5 primitive endoderm by the KeyGenes algorithm (Fig. 2e) and they express canonical primitive endoderm markers (Supplementary Fig. 4d). Hence, we are convinced that *Pdgfra* is in fact a good marker for XEN-like cells.

4. The authors observed that continuous treatment with RA promotes appearance of XEN cells. They also identify groups of transcription factors that might be involved in the cell identity specification. The obvious question is whether RA treatment influences expression of any of these putative “patterning” factors.

Tbx3 and likely also *Gbx2* are direct targets of RA. We now state that explicitly together with relevant references in the new manuscript.

Additional question is that differentiation without RA seems to be much more homogeneous towards neuroectodermal lineage. It would be interesting to examine whether these cells do not undergo similar stage of “Expression noise” as cells exposed to RA.

We agree that a comparison with differentiation without RA would be interesting. However, for a rigorous comparison a comprehensive single-cell RNA-seq measurement of differentiation without RA would be necessary, which we feel is outside of the scope of this manuscript. Nevertheless, we have collected new smFISH data on *Sox2* at 24 h of differentiation with and without RA. These data show that *Sox2* expressions is bimodal in the presence of RA, which indicates the lineage bifurcation. In the absence of RA, on the other hand, *Sox2* expression is unimodal, which indicates a more homogeneous population. We believe that this is not a main point of our paper and therefore did not include this data. However, the data could be included, if deemed necessary.

5. For these transcription factor analysis (Fig. 4), Supplementary Methods describe that the authors used only 39 out of 82 cells (page 15 in Supplementary Methods, “Out of the 82 cells measured by SMART-seq at 48 h, 10 were considered XEN-like and 29 ectoderm-like”). The authors may find other groups of transcription factors from the rest of cells (43 out 82 cells), or at least need to explain what expression patterns for these 43 of the filtered cells look like.

We have now also considered the remaining cells. This population of cells expresses several pluripotency markers (*Zfp42*, *Stat1*) and is thus likely a mixture of undifferentiated cells and epiblast-like cells. Our findings are reported in the main manuscript text and Supplementary Fig. 9b.

6. Substantial amount of results in the paper are shown in the Supplementary Figures. However, Some of Supplementary Figures in this paper are not ordered properly, not matched with figure contents, or nothing mentioned in the paper although some figures present. The following are some of examples. The authors should arrange and explain carefully Supplementary Figures according to the results in the this paper.

- a. In page 4 in the text, Supplementary Figs. 3a and 3b should show that “Cell type frequencies were quantified after 96 h using antibody staining”. However, Supplementary Figs. 3a and 3b do not show these results (Antibody staining in Supplementary Figs. 3a and 3b is missing).**
- b. “Bulk RNA-seq” clustering analyses shown in Supplementary Figs. 3a and 3b is not mentioned in the paper.**
- c. Supplementary figs. 2a-e and 2g-i are not mentioned in the main text or figure legends.**
- d. Supplementary Figs. 4a and 4b are not mentioned in the main text, figure legends, or Supplementary Methods.**

e. Supplementary Figs. 8a and 8b related to Fig. 4 (page 23) are missing and not shown anywhere in the paper.

f. In the figure legend for Supplementary Fig. 3, “d, Graphical representation of the data in e”. However, “the data in e” is unclear. Does e indicate Fig. 2e or 3e or missing?

We would like to thank the reviewer for alerting us to these mistakes. All supplementary figures are now mentioned in the main text. In the case of Supplementary Fig. 2 we do not refer to specific panels, since the technical specifications of the method are given in the Methods section. Since the method is not the focus of this manuscript and has been described elsewhere, we feel that a detailed technical discussion of its performance would distract from the main thread of the paper.

Reviewer #3 (Remarks to the Author):

I suspect that many pluripotency student will be very surprised to see XEN appear as an alternative fate in the exit of pluripotency. Most of the works on this subject consider an alternative between ectoderm (or neuroectoderm) and mesendoderm and therefore it will surprise them to see the fate choice revealed by the analysis. The authors should comment on this and make a reference to the plausibility of the observation. From the perspective of this reviewer, the observation is sound. The time course of the differentiation, the quality of the data, the consistency of the results, all point to the fact that this is a genuine observation. Why then is there no trace of this fate decision in the literature? It will be interesting and significant that the authors discussed this matter.

I would suggest that they notice that they start from cells grown in 2i+LIF and that it has been suggested that long term exposure to this medium can turn ES cells into quasi-totipotent cells (Morgani et al. Totipotent embryonic stem cells arise in ground-state culture conditions. Cell Rep. 2013 Jun 27;3(6):1945-57). Could it therefore be that, in their experiments what they are seeing is a fate choice from an ICM like population in which cells choose between a XEN fate and an advanced –RA induced- epiblast fate? It would be really helpful to readers and, also to the authors, to comment on the unusual nature of this observation and some possible explanations. In this regard, there is clearly a hint of neural fate in the XEN population but, in the face of what is shown, it is not that clearly neural; could it be a modified epiblast? Have they looked at epiblast markers such as Fgf5, Otx2, Pou3f1, 2? Can they rule out that they are not seeing an EPI v XEN decisions?

We are thankful to the reviewer for raising this issue. Due to this remark and a similar remark from reviewer 1 we have re-examined our data and found clear indications that the initial lineage decision is in fact between epiblast and XEN. Both by pseudotemporal ordering of our single-cell RNA-seq data set (Supplementary Fig. 6) and analysis of 12

new bulk RNA-seq data sets (Fig. 2d-e) we revealed the transient expression of early epiblast-markers. We found that PDGFRA negative/CD24 high cells expressed early epiblast markers at 48 h and 72 h and were identified as E5.5 epiblast by the KeyGenes algorithm (Roost et al., Stem Cell Rep. 2016). We believe that epiblast-like cells quickly differentiate further to neuroectoderm, due to continuous exposure to RA. In the new manuscript we support this claim by two new bulk RNA-seq data sets of purified ectoderm- or XEN like cells after continued culture (Supplementary Fig. 4d-f). We found that the two cell lines expressed markers for neuroectoderm/neural crest or primitive endoderm, respectively, and had expression profiles similar to corresponding tissues *in vivo*. All in all, we believe that we have now a more complete picture of the lineage bifurcation, which seems to be more in line with what one would expect from *in vivo* development.

In the argument raised above an important issue is that the experiments are started from 2i+LIF, thus something that would be of interest is to see what happens if the same experiment was started from LIF + BMP or Serum + LIF. I am raising this as a thought experiment and not as something the authors should do, though they may have some relevant results.

We followed the reviewer's suggestion and carried out RA differentiation after culture in serum + LIF. Interestingly, under these conditions cells lost the ability to differentiate to XEN-like cells while ectoderm-like cells were still present (Supplementary Fig. 7a-b). This new experiment supports a model in which cells cultured in serum+LIF functionally correspond to a slightly later developmental time point than cells grown in 2i+LIF. We are grateful to the reviewer for suggesting this useful experiment.

These matters aside, there are a number of small issues: The authors use Martello et al (ref 30) as their reference for pluripotency network. There are others and there is no reason, to quote this exclusively.

We added additional references describing the pluripotency network.

In page 4, the discussion of the experiments on the effect of MEKi on RA induction could and should be explained better.

In the revised manuscript we additionally studied the effect of an FGF receptor inhibitor and contrast these new experiments with the effect of MEKi. We further focused the stochastic simulations on the difference between RA delay and an RA pulse (or timed MEKi exposure). Further, we significantly extended the discussion of the signaling experiments and placed them in context of recently published *in vivo* results (Saiz et al., Nature Comm., 2016).

In page 5, the authors talk about a phase in which cells coexpress XEN and

neuroectoderm markers; it would be helpful if they could provide some examples of this in the main text. This is an important point and should be illustrated properly.

We do agree with the reviewer that the co-expression of XEN- and neuroectoderm-specific early regulators is an interesting finding. In fact, we do discuss an example (*Tbx3* and *Gbx2*) in detail. However, since live cell tracking experiments would be necessary to rigorously show a lineage biasing function of those regulators, we prefer not to emphasize this finding more at this point.

In the discussion, the authors mention the possibility that some pluripotency genes might be involved in differentiation and make two references 31 and 32. They might want to also include Malleshaiah et al. Nac1 Coordinates a Sub-network of Pluripotency Factors to Regulate Embryonic Stem Cell Differentiation. Cell Rep. 2016 Feb 9;14(5):1181-94, which shows some relevant data and present some ideas and models related to those under discussion here.

We included the suggested reference in the Discussion section.

In Supplementary Fig 3: d it says: says d, Graphical representation of the data in e. It probably means 'in c' rather than 'in e'.

We corrected this mistake.

In Supplementary Fig6B, some of the proteins listed as TFs are NOT TFs e.g axin2, fgfr2, tsc22d4, dedd2, fgf10, maybe there are more. The authors should make sure that if they are talking about TFs what they refer to are TFs.

Indeed, not all of the considered genes are transcription factors, but all supposedly impact transcription. In the revised version of the manuscript we call the considered genes “transcriptional regulators” and refer explicitly to the GO terms we used to define the set. We hope that this wording is acceptable.

Reviewers' comments:

Reviewer #1 (Remarks to the Author):

The authors have made a generally reasonable response to suggestions and criticisms in their formal response to reviewers' statement. Unfortunately, however, this is not well reflected in the revised manuscript. Rather than re-consider their argument as a whole, the authors have simply shoe-horned in a few additional paragraphs and pieces of data resulting in an incoherent patchwork of contradictory statements. I do not find the manuscript acceptable for publication with the current text due to errors and inconsistencies. This is a missed opportunity because they provide a rich and timely dataset and the findings would be of wide interest if presented in a correct developmental biology framework. I would strongly encourage the authors to reconsider their results from first principles and provide a more accurate discourse that is better integrated with prior knowledge.

There are two major issues that require explicit clarification. The first is the "lineage bifurcation" that the authors continue to insist on from the Abstract onwards. The second, which becomes confused with the first, is the definition of epiblast. As the authors accept in their response statement, there is no bifurcation between primitive endoderm and ectoderm in the embryo. They now invoke a bifurcation between primitive endoderm and epiblast, while at the same time stating that ES cells already have an epiblast identity (p5 "Keygenes identified mESCs as E4.5 epiblast, in agreement with previous results"). It is not logical to say that ES cells represent E4.5 epiblast, which indeed is consistent with both classical and current studies, and at same time that they "mimic the epiblast/primitive endoderm lineage decision that occurs in vivo". What is the decision if the cells are already epiblast? ES cells may indeed be capable of making primitive endoderm in certain culture conditions, but the developmental basis for this is unclear and I agree with reviewer 3 that the authors need to discuss the complexity of this issue rather than over-simplify. There are various possibilities. For example, a sub-population of ES cells might represent an earlier stage of ICM cells prior to the epiblast/primitive endoderm segregation. The present authors could use their single cell data to evaluate this proposition. A second possibility would be that RA stochastically induces a de- or trans-differentiation phenomenon in a fraction of ES cells. This may not be unreasonable given that the majority of pluripotency factors (not just Tbx3) are retained in primitive endoderm during initial lineage segregation (see for example, Boroviak et al, 2015). Neither of these processes seem to meet criteria for true bifurcations, by which I take the authors to mean a binary lineage choice at the single cell level. A third possibility, which would be more in line with a bifurcation, is that the culture conditions may provoke a lineage decision between epiblast progression (see below) and primitive endoderm. In such case, it has to be acknowledged that this would be an in vitro phenomenon. That does not necessarily mean less interesting; indeed one might even argue it is intrinsically extremely interesting if artificial lineage choices can be engineered. The authors' datasets could likely shed light on which process is in play, but minimally they must at least discuss the possibilities rather than ignore text book and contemporary understanding of mouse embryo development. In that vein, epiblast in embryology describes a pluripotent tissue formed following segregation of the primitive endoderm that persists until the onset of somitogenesis. The epiblast changes progressively in the embryo (see for example Hayashi et al, 2011; Acampora et al, 2016) and therefore the authors must be more precise about the stages of epiblast they are referring to and resolve the textual contradictions on p5. It is also not helpful in this context to describe epiblast as a "default lineage", or for that matter any lineage as "default" – what does that mean?

There are also some technical issues that require correction:

1. LIF is not a "component of the defined 2i medium" as stated on p6 and implied in the methods. It is an optional addition and ES cells cultured in 2i are somewhat different, molecularly and functionally, from ES cells cultured in 2i plus LIF (Wray et al, 2010; Dunn et al, 2014). The authors should be specific as to whether they are culturing in 2i or in 2i plus LIF. They also indicate in the methods that titrated doses on the two inhibitors are used at some points, but I do not find where indicated in the text or figure legends.
2. "RA driven XEN-specification involves the same signalling pathways". "Involves" should be

requires because there is no evidence that the signalling pathways are downstream of RA, rather they operate in parallel.

3. "In agreement with the earliest report on N2B27 differentiation we consistently found only ectoderm-like cells when there was no RA present or if the MEK inhibitor was applied from the beginning (Fig 3a-c)". This statement is wrong on two counts. Firstly previous reports, including the one cited, have described a proportion on non-neuroectoderm cells in the N2B27 differentiation system (Ying et al, 2003; Lowell et al, 2006; Malaguti et al, 2013; Kalkan et al, 2017). Second, the authors' own data in Fig 3b clearly show the presence of non-ectoderm (PDGFRA-/CD24-) cells and that these actually become the majority in the continuous presence of MEK inhibitor, with a 2-fold reduction in the frequency of ectoderm cells (contrary to statements in the text).

4. The FGF receptor inhibitor is used at 1 μ M. This is a ten-fold excess concentration which will have growth inhibitory and toxic effects, likely contributing to the difference in response compared with MEKi.

5. "Saiz et al further revealed that MEKi prevented the specification of primitive endoderm". In fact this was revealed several years earlier by Nichols et al (Development, 2009).

Reviewer #2 (Remarks to the Author):

the authors satisfactorily addressed the key questions.

Reviewer #3 (Remarks to the Author):

This is a revised version of what was, already, an interesting and important piece of work. However, the work contained some results that could be deemed controversial, particularly concerning the XEN lineage emerging from a differentiation protocol of a pluripotent ES cell populations (and the other referees have identified this as well). The authors have clarified this issue beyond reasonable doubt and, in doing so, have established important observations for the field about the state of ESCs in 2i. Interestingly there are observations deemed anecdotal in the literature in support of this observation (e.g ref 41) and the experiments performed here go a long way to provide a sound basis for believing those observations. It would be helpful if the authors could also quote the work of Schroeter et al (PMID: 26511924 which shows that exposure to 2i increases the probability of differentiation into XEN).

The authors have added much data which turns the paper into the most thorough work I know of on the differentiation of mouse ES cells. The number and volume of new experiments and data is much appreciated. The discussion of the referees comments is insightful and helpful and has been incorporated into the manuscript.

Just a few small points that the authors might want to incorporate into the manuscript. At the beginning they seem to consider 96hrs as a short differentiation time when it is actually 4 days! They might want to remove the 'only'.

The apparent increase in variability at the exit of pluripotency echoes recent reports on similar observations and the authors might want to comment on this as, perhaps, a feature of cell fate decisions and cite those papers (PMID: 28027308 and 28027290)

The notion of a default neural state is an important one brought to light here, A bit of an urban legend which gets some support here and the authors might want to comment on that. Their model is very good but, on the issue of default neural they might want to quote ref 54, which also deals with the same issue in more primitive but not dissimilar manner. I would understand if the authors do not think the reference is merited.

All in all an excellent piece of work.

Please find below our detailed responses to the reviewers' concerns. In the manuscript all sections that are substantially new are highlighted in blue.

Reviewer #1 (Remarks to the Author):

There are two major issues that require explicit clarification. The first is the “lineage bifurcation” that the authors continue to insist on from the Abstract onwards. The second, which becomes confused with the first, is the definition of epiblast. As the authors accept in their response statement, there is no bifurcation between primitive endoderm and ectoderm in the embryo. They now invoke a bifurcation between primitive endoderm and epiblast, while at the same time stating that ES cells already have an epiblast identity (p5 “Keygenes identified mESCs as E4.5 epiblast, in agreement with previous results”). It is not logical to say that ES cells represent E4.5 epiblast, which indeed is consistent with both classical and current studies, and at same time that they “mimic the epiblast/primitive endoderm lineage decision that occurs in vivo”. What is the decision if the cells are already epiblast?

We agree with the reviewer that our language was imprecise. In this version of the manuscript we addressed both points to avoid any confusion. Indeed, we found that mESCs in 2i+LIF are most similar to E4.5 epiblast, in agreement with previous results. Hence, the gene expression dynamics observed by us *in vitro* has no direct *in vivo* equivalent.

In the current version of the manuscript we have removed the statement “mimicking the epiblast/primitive endoderm lineage decision that occurs in vivo” and discriminated more carefully between different stages of epiblast development. Throughout the manuscript it is now clearly stated that the decision occurs between progression along the epiblast lineage or trans-differentiation to primitive endoderm (see also our reply to the next point).

ES cells may indeed be capable of making primitive endoderm in certain culture conditions, but the developmental basis for this is unclear and I agree with reviewer 3 that the authors need to discuss the complexity of this issue rather than over-simplify. There are various possibilities. For example, a sub-population of ES cells might represent an earlier stage of ICM cells prior to the epiblast/primitive endoderm segregation. The present authors could use their single cell data to evaluate this proposition. A second possibility would be that RA stochastically induces a de- or trans-differentiation phenomenon in a fraction of ES cells. This may not be unreasonable given that the majority of pluripotency factors (not just Tbx3) are retained in primitive endoderm during initial lineage segregation (see for example, Boroviak et al, 2015). Neither of these processes seem to meet criteria for true bifurcations, by which I take the authors to mean a binary lineage choice at the single cell level. A third possibility, which would be

more in line with a bifurcation, is that the culture conditions may provoke a lineage decision between epiblast progression (see below) and primitive endoderm. In such case, it has to be acknowledged that this would be an *in vitro* phenomenon. That does not necessarily mean less interesting; indeed one might even argue it is intrinsically extremely interesting if artificial lineage choices can be engineered. The authors' datasets could likely shed light on which process is in play, but minimally they must at least discuss the possibilities rather than ignore text book and contemporary understanding of mouse embryo development.

To discriminate between the different biological possibilities suggested by the reviewer, which could underlie the potential of ESCs to make primitive endoderm, we compared our single-cell data set with the *in vivo* data from Boroviak et al. 2015 (see new Fig. 4a and new Fig. S6). This analysis showed that, under RA, cells first moved towards post-implantation (E5.5) epiblast prior the occurrence of two separate subpopulations. The subpopulation we identified as XEN-like was closest to the E4.5 primitive endoderm tissue measured by Boroviak et al. This analysis thus demonstrates that RA differentiation does not follow the dynamics of *in vivo* development. Instead, XEN-like cells are likely generated through trans-differentiation from E4.5 or E5.5 epiblast-like cells. We ensured that this observation and the similarities as well as differences between *in vivo* and *in vitro* are clearly stated in the manuscript now.

In that vein, epiblast in embryology describes a pluripotent tissue formed following segregation of the primitive endoderm that persists until the onset of somitogenesis. The epiblast changes progressively in the embryo (see for example Hayashi et al, 2011; Acampora et al, 2016) and therefore the authors must be more precise about the stages of epiblast they are referring to and resolve the textual contradictions on p5.

In the current version of the manuscript we explicitly discriminate between pre-implantation (E4.5) and post-implantation (E5.5) epiblast.

It is also not helpful in this context to describe epiblast as a “default lineage”, or for that matter any lineage as “default” – what does that mean?

We called epiblast the default lineage as previously done (see, for example, Saiz et al., Nature Comm., 2016) since *in vivo*, ICM cells become epiblast in the absence of primitive endoderm inducing signals. However, to avoid any confusion we no longer make use of the term “default”.

There are also some technical issues that require correction:

- 1. LIF is not a “component of the defined 2i medium” as stated on p6 and implied in the methods. It is an optional addition and ES cells cultured in 2i**

are somewhat different, molecularly and functionally, from ES cells cultured in 2i plus LIF (Wray et al, 2010; Dunn et al, 2014). The authors should be specific as to whether they are culturing in 2i or in 2i plus LIF. They also indicate in the methods that titrated doses on the two inhibitors are used at some points, but I do not find where indicated in the text or figure legends.

Both of these comments have been addressed. We now state explicitly that we use 2i medium plus LIF (2i/L). The ranges of the GSK3 and MEK inhibitor used for the differentiation experiments are mentioned in the legend of Fig. S7.

2. “RA driven XEN-specification involves the same signalling pathways”. “Involves” should be requires because there is no evidence that the signalling pathways are downstream of RA, rather they operate in parallel.

The reviewer is correct. We changed “involves” to “requires”.

3. “In agreement with the earliest report on N2B27 differentiation we consistently found only ectoderm-like cells when there was no RA present or if the MEK inhibitor was applied from the beginning (Fig 3a-c)”. This statement is wrong on two counts. Firstly previous reports, including the one cited, have described a proportion on non-neuroectoderm cells in the N2B27 differentiation system (Ying et al, 2003; Lowell et al, 2006; Malaguti et al, 2013; Kalkan et al, 2017). Second, the authors’ own data in Fig 3b clearly show the presence of non-ectoderm (PDGFRA-/CD24-) cells and that these actually become the majority in the continuous presence of MEK inhibitor, with a 2-fold reduction in the frequency of ectoderm cells (contrary to statements in the text).

The reviewer is correct in pointing out that not all cells become ectoderm-like. We now clearly state that the “majority” of cells (and not all) become ectoderm-like in the absence of RA, which reflects our measurements and is consistent with previous reports.

4. The FGF receptor inhibitor is used at 1 μ M. This is a ten-fold excess concentration which will have growth inhibitory and toxic effects, likely contributing to the difference in response compared with MEKi.

We have not observed increased cell death or reduced growth in our experiments with the FGF receptor inhibitor PD173074. This inhibitor has been used previously at the same concentration (Schroeter et al., Development, 2015) but no growth inhibitory or cytotoxic effects have been reported there. Another study reported less than 10%

growth inhibition when 1 μ M PD173074 was used (Anreddy et al., Acta Pharm Sinica B, 2014). These previous reports as well as our own observation makes it unlikely that cytotoxic or growth inhibitory effects underlie the observed differences between PD173074 and MEKi.

5. “Saiz et al further revealed that MEKi prevented the specification of primitive endoderm”. In fact this was revealed several years earlier by Nichols et al (Development, 2009).

We thank the reviewer for pointing out this omission. In the current version of the manuscript we cite Nichols et al.

Reviewer #3 (Remarks to the Author):

It would be helpful if the authors could also quote the work of Schroeter et al (PMID: 26511924 which shows that exposure to 2i increases the probability of differentiation into XEN.

We agree and in the current version of the manuscript we additionally mention explicitly the relevant result from that study.

At the beginning they seem to consider 96 hrs as a short differentiation time when it is actually 4 days! They might want to remove the ‘only’.

We agree. We removed the word “only”.

The apparent increase in variability at the exit of pluripotency echoes recent reports on similar observations and the authors might want to comment on this as, perhaps, a feature of cell fate decisions and cite those papers (PMID: 28027308 and 28027290)

We thank the reviewer for pointing us to these studies. The increase in variability at the exit from pluripotency might very likely be a general phenomenon. We have included the suggested references in the discussion.

The notion of a default neural state is an important one brought to light here, A bit of an urban legend which gets some support here and the authors might want to comment on that. Their model is very good but, on the issue of default neural they might want to quote ref 54, which also deals with the same issue in more primitive but not dissimilar manner. I would understand if the authors do not think the reference is merited.

In the current version of the manuscript we have included the suggested reference in the discussion of our model. We now also state explicitly that our model does not strictly require an ectoderm bias. An initial bias towards progression along the epiblast lineage is sufficient to explain the observed differences between RA pulse and delay regimens. An epiblast bias is thus in line with the observations by Trott et. al.

REVIEWERS' COMMENTS:

Reviewer #1 (Remarks to the Author):

The narrative of the revised manuscript is improved in clarity and accuracy. Notably the authors acknowledge the inconsistency between their in vitro lineage diversification and developmental chronology in the embryo and discuss possible reasons for this. Overall the paper complements and extends other recent investigations of mouse ES cell progression to lineage commitment and the single cell transcriptome data will be a useful resource for the community. Consideration of expression variability and noise during cell decision making is topical. Although the GRN simulation appears constructed to fit the authors' assumptions rather than emerging from the data, it may provoke more formal analyses. I support publication.

Reviewer #1 (Remarks to the Author):

The narrative of the revised manuscript is improved in clarity and accuracy. Notably the authors acknowledge the inconsistency between their in vitro lineage diversification and developmental chronology in the embryo and discuss possible reasons for this. Overall the paper complements and extends other recent investigations of mouse ES cell progression to lineage commitment and the single cell transcriptome data will be a useful resource for the community. Consideration of expression variability and noise during cell decision making is topical. Although the GRN simulation appears constructed to fit the authors' assumptions rather than emerging from the data, it may provoke more formal analyses. I support publication.

We thank the reviewer for their productive criticism in previous rounds of the revision process and are pleased that we could convince them of the merit of our work. The reviewer points out correctly that the GRN model was not derived from the data but is an adaptation of a model that has been previously used to describe lineage bifurcations. Nevertheless, this very simple model correctly recapitulates the observed lineage response dynamics and thus provides an intuitive way to gain insight into the differentiation process.